# Rectangular Flows for Manifold Learning

**Anthony L. Caterini**[*]
University of Oxford & Layer 6 AI
anthony@layer6.ai

**Gabriel Loaiza-Ganem**[*]
Layer 6 AI
gabriel@layer6.ai

**Geoff Pleiss**
Columbia University
gmp2162@columbia.edu

**John P. Cunningham**
Columbia University
jpc2181@columbia.edu

## Abstract

Normalizing flows are invertible neural networks with tractable change-of-volume terms, which allow optimization of their parameters to be efficiently performed via maximum likelihood. However, data of interest are typically assumed to live in some (often unknown) low-dimensional manifold embedded in a high-dimensional ambient space. The result is a modelling mismatch since – by construction – the invertibility requirement implies high-dimensional support of the learned distribution. Injective flows, mappings from low- to high-dimensional spaces, aim to fix this discrepancy by learning distributions on manifolds, but the resulting volume-change term becomes more challenging to evaluate. Current approaches either avoid computing this term entirely using various heuristics, or assume the manifold is known beforehand and therefore are not widely applicable. Instead, we propose two methods to tractably calculate the gradient of this term with respect to the parameters of the model, relying on careful use of automatic differentiation and techniques from numerical linear algebra. Both approaches perform end-to-end nonlinear manifold learning and density estimation for data projected onto this manifold. We study the trade-offs between our proposed methods, empirically verify that we outperform approaches ignoring the volume-change term by more accurately learning manifolds and the corresponding distributions on them, and show promising results on out-of-distribution detection. Our code is available at https://github.com/layer6ai-labs/rectangular-flows.

## 1 Introduction

In recent years, Normalizing Flows (NFs) have become a staple of generative modelling, being widely used for density estimation [14, 15, 45, 28, 16], variational inference [52, 30], maximum entropy modelling [37], and more [46, 31]. In density estimation, we typically have access to a set of points living in some high-dimensional space $\mathbb{R}^D$. NFs model the corresponding data-generating distribution as the pushforward of a simple distribution on $\mathbb{R}^D$ – often a Gaussian – through a smooth bijective mapping. Clever construction of these bijections allows for tractable density evaluation and thus maximum likelihood estimation of the parameters. However, as an immediate consequence of this choice, the learned distribution has support homeomorphic to $\mathbb{R}^D$; in particular, the resulting distribution is supported on a set of dimension $D$. This is not a realistic assumption in practice – especially for density estimation – as it directly contradicts the manifold hypothesis [6] which states that high-dimensional data lives on a lower-dimensional manifold embedded in ambient space.

---

[*]Authors contributed equally.

35th Conference on Neural Information Processing Systems (NeurIPS 2021).

A natural idea to circumvent this misspecification is to consider *injective* instead of bijective flows, which now push forward a random variable on $\mathbb{R}^d$ with $d < D$ to obtain a distribution on some $d$-dimensional manifold embedded in $\mathbb{R}^D$. These mappings admit a change-of-variable formula bearing resemblance to that of bijective flows, but unfortunately the volume-change term becomes computationally prohibitive, which then impacts the tractability of maximum likelihood. While there have been recent efforts towards training flows where the resulting distribution is supported on a low-dimensional manifold [18, 53, 8, 35, 40, 12], these approaches either assume that the manifold is known beforehand or propose various heuristics to avoid the change-of-variable computation. Both of these are undesirable, because, while we should expect most high-dimensional data of interest to exhibit low-dimensional structure, this structure is almost always unknown. On the other hand, we argue that avoiding the volume-change term may result in learning a manifold to which it is difficult to properly assign density, and this approach further results in methods which do not take advantage of density evaluation, undermining the main motivation for using NFs in the first place.

We show that density estimation for injective flows based on maximum likelihood can be made tractable. By carefully leveraging forward- and backward-mode automatic differentiation [3], we propose two methods that allow backpropagating through the volume term arising from the injective change-of-variable formula. The first method involves exact evaluation of this term and its gradient which incurs a higher memory cost; the second uses conjugate gradients [43] and Hutchinson's trace estimator [23] to obtain unbiased stochastic gradient estimates. Unlike previous work, our methods do not need the data manifold to be specified beforehand, but instead simultaneously estimate this manifold along with the distribution on it end-to-end, thus enabling maximum likelihood training to occur. To the best of our knowledge, ours are the first methods to scale backpropagation through the injective volume-change term to ambient dimensions $D$ close to 3,000. We study the trade-off between memory and variance introduced by our methods and show empirical improvements over injective flow baselines for density estimation. We also show that injective flows obtain state-of-the-art performance for likelihood-based Out-of-Distribution (OoD) detection, assigning higher likelihoods to Fashion-MNIST (FMNIST) [57] than to MNIST [36] with a model trained on the former.

## 2 Background

### 2.1 Square Normalizing Flows

A normalizing flow [52, 15] is a diffeomorphism $\tilde{f}_\theta : \mathbb{R}^D \to \mathbb{R}^D$ parametrized by $\theta$, that is, a differentiable bijection with differentiable inverse. Starting with a random variable $Z \sim p_Z$ for a simple density $p_Z$ supported on $\mathbb{R}^D$, e.g. a standard Gaussian, the change-of-variable formula states that the random variable $X := \tilde{f}_\theta(Z)$ has density $p_X$ on $\mathbb{R}^D$ given by:

$$p_X(x) = p_Z\left(\tilde{f}_\theta^{-1}(x)\right) \left|\det \mathbf{J}\left[\tilde{f}_\theta\right]\left(\tilde{f}_\theta^{-1}(x)\right)\right|^{-1}, \tag{1}$$

where $\mathbf{J}[\cdot]$ is the differentiation operator, so that $\mathbf{J}[\tilde{f}_\theta](\tilde{f}_\theta^{-1}(x)) \in \mathbb{R}^{D \times D}$ is the Jacobian of $\tilde{f}_\theta$ (with respect to the inputs and not $\theta$) evaluated at $\tilde{f}_\theta^{-1}(x)$. We refer to this now standard setup as *square flows* since the Jacobian is a square matrix. The change-of-variable formula is often written in terms of the Jacobian of $\tilde{f}_\theta^{-1}$, but we use the form of (1) as it is more applicable for the next section. NFs are typically constructed in such a way that not only ensures bijectivity, but also so that the Jacobian determinant in (1) can be efficiently evaluated. When provided with a dataset $\{x_i\}_{i=1}^n \subset \mathbb{R}^D$, an NF models its generating distribution as the pushforward of $p_Z$ through $\tilde{f}_\theta$, and thus the parameters can be estimated via maximum likelihood as $\theta^* := \arg\max_\theta \sum_{i=1}^n \log p_X(x_i)$.

### 2.2 Rectangular Normalizing Flows

As previously mentioned, square NFs unrealistically result in the learned density $p_X$ having $D$-dimensional support. We follow the injective flow construction of Brehmer and Cranmer [8], where a smooth and injective mapping $g_\phi : \mathbb{R}^d \to \mathbb{R}^D$ with $d < D$ is constructed. In this setting, $Z \in \mathbb{R}^d$ is the low-dimensional variable used to model the data as $X := g_\phi(Z)$. A well-known result from differential geometry [32] provides an applicable change-of-variable formula:

$$p_X(x) = p_Z\left(g_\phi^{-1}(x)\right) \left|\det \mathbf{J}[g_\phi]^\top \left(g_\phi^{-1}(x)\right) \mathbf{J}[g_\phi]\left(g_\phi^{-1}(x)\right)\right|^{-1/2} \mathbb{1}(x \in \mathcal{M}_\phi), \tag{2}$$

where $\mathcal{M}_\phi := \{g_\phi(z) : z \in \mathbb{R}^d\}$. The Jacobian-transpose-Jacobian determinant now characterizes the change in volume from $Z$ to $X$. We make several relevant observations: $(i)$ The Jacobian matrix $\mathbf{J}[g_\phi](g_\phi^{-1}(x)) \in \mathbb{R}^{D \times d}$ is no longer a square matrix, and we thus refer to these flows as *rectangular*. $(ii)$ Note that $g_\phi^{-1} : \mathcal{M}_\phi \to \mathbb{R}^d$ is only properly defined on $\mathcal{M}_\phi$ and not $\mathbb{R}^D$, and $p_X$ is now supported on the $d$-dimensional manifold $\mathcal{M}_\phi$. $(iii)$ We write the indicator $\mathbb{1}(x \in \mathcal{M}_\phi)$ explicitly to highlight the fact that this density is *not* a density with respect to the Lebesgue measure; rather, the dominating measure is a Riemannian measure on the manifold $\mathcal{M}_\phi$ [48]. $(iv)$ One can clearly verify as a sanity check that when $d = D$, equation (2) reduces to (1).

Since data points $x$ will almost surely not lie exactly on $\mathcal{M}_\phi$, we use a left inverse $g_\phi^\dagger : \mathbb{R}^D \to \mathbb{R}^d$ in place of $g_\phi^{-1}$ such that $g_\phi^\dagger(g_\phi(z)) = z$ for all $z \in \mathbb{R}^d$, which exists because $g_\phi$ is injective. This is properly defined on $\mathbb{R}^D$, unlike $g_\phi^{-1}$ which only exists over $\mathcal{M}_\phi$. Equation (2) then becomes:

$$p_X(x) = p_Z\left(g_\phi^\dagger(x)\right) \left|\det \mathbf{J}[g_\phi]^\top \left(g_\phi^\dagger(x)\right) \mathbf{J}[g_\phi] \left(g_\phi^\dagger(x)\right)\right|^{-1/2}. \tag{3}$$

Note that (3) is equivalent to projecting $x$ onto $\mathcal{M}_\phi$ as $x \leftarrow g_\phi(g_\phi^\dagger(x))$, and then evaluating the density from (2) at the projected point.

Now, $g_\phi$ is injectively constructed as follows:

$$g_\phi = \tilde{f}_\theta \circ \texttt{pad} \circ h_\eta \qquad \text{and} \qquad g_\phi^\dagger = h_\eta^{-1} \circ \texttt{pad}^\dagger \circ \tilde{f}_\theta^{-1}, \tag{4}$$

where $\tilde{f}_\theta : \mathbb{R}^D \to \mathbb{R}^D$ and $h_\eta : \mathbb{R}^d \to \mathbb{R}^d$ are both square flows, $\phi := (\theta, \eta)$, and $\texttt{pad} : \mathbb{R}^d \to \mathbb{R}^D$ and $\texttt{pad}^\dagger : \mathbb{R}^D \to \mathbb{R}^d$ are defined as $\texttt{pad}(z) = (z, \mathbf{0})$ and $\texttt{pad}^\dagger(z, z') = z$, where $\mathbf{0}, z' \in \mathbb{R}^{D-d}$. Now, $\mathcal{M}_\phi$ depends only on $\theta$ and not $\eta$, so we write it as $\mathcal{M}_\theta$ from now on. Applying (3) yields:

$$p_X(x) = p_Z\left(g_\phi^\dagger(x)\right) \left|\det \mathbf{J}[h_\eta] \left(g_\phi^\dagger(x)\right)\right|^{-1} \left|\det \mathbf{J}[f_\theta]^\top \left(f_\theta^\dagger(x)\right) \mathbf{J}[f_\theta] \left(f_\theta^\dagger(x)\right)\right|^{-1/2}, \tag{5}$$

where $f_\theta = \tilde{f}_\theta \circ \texttt{pad}$ and $f_\theta^\dagger = \texttt{pad}^\dagger \circ \tilde{f}_\theta^{-1}$. We include a derivation of (5) in Appendix A, along with a note on why injective transformations cannot be stacked as naturally as bijective ones.

Evaluating likelihoods is seemingly intractable since constructing flows with a closed-form volume-change term is significantly more challenging than in the square case, even if the relevant matrix is now $d \times d$ instead of $D \times D$. Brehmer and Cranmer [8] thus propose a two-step training procedure to promote tractability wherein $f_\theta$ and $h_\eta$ are trained separately. After observing that there is no term encouraging $x \in \mathcal{M}_\theta$, and that $x \in \mathcal{M}_\theta \iff x = g_\phi(g_\phi^\dagger(x)) \iff x = f_\theta(f_\theta^\dagger(x))$, they decide to simply train $f_\theta$ by minimizing the reconstruction error to encourage the observed data to lie on $\mathcal{M}_\theta$:

$$\theta^* = \arg\min_\theta \sum_{i=1}^n \left|\left|x_i - f_\theta\left(f_\theta^\dagger(x_i)\right)\right|\right|_2^2. \tag{6}$$

Note that the above requires computing both $f_\theta$ and $f_\theta^\dagger$, so that $\tilde{f}_\theta$ should be chosen as a flow allowing fast evaluation of both $\tilde{f}_\theta$ and $\tilde{f}_\theta^{-1}$. Architectures such as the Real NVP [15] or follow-up work [28, 16] are thus natural choices for $\tilde{f}_\theta$, while architectures with an autoregressive component [45, 30] should be avoided. Then, since $h_\eta$ does not appear in the challenging determinant term in (5), $h_\eta$ can be chosen as any normalizing flow, and optimization – for a fixed $\theta$ – can be tractably achieved by maximum likelihood over the lower-dimensional space:

$$\eta^* = \arg\max_\eta \sum_{i=1}^n \left\{\log p_Z\left(g_\phi^\dagger(x_i)\right) - \log\left|\det \mathbf{J}[h_\eta]\left(g_\phi^\dagger(x_i)\right)\right|\right\}. \tag{7}$$

In practice, gradient steps in $\theta$ and $\eta$ are alternated. This entire procedure circumvents evaluation of the Jacobian-transpose-Jacobian determinant term in (5), but as we show in section 3, avoiding this term by separately learning the manifold and the density on it comes with its downsides. We then show how to tractably estimate this term in section 4.

## 3   Related Work and Motivation

**Low-dimensional and topological pathologies**   The mismatch between the dimension of the modelled support and that of the data-generating distribution has been observed throughout the literature in various ways. Dai and Wipf [13] show, in the context of variational autoencoders [29], that using flexible distributional approximators supported on $\mathbb{R}^D$ to model data living in a low-dimensional manifold results in pathological behavior where the manifold itself is learned, but not the distribution on it. Cornish et al. [11] demonstrate the drawbacks of normalizing flows for estimating the density of topologically-complex data, and provide a new numerically stable method for learning NFs when the support is not homeomorphic to $\mathbb{R}^D$. However, this approach still models the support as being $D$-dimensional. Behrmann et al. [5] show instabilities associated with NFs – particularly a lack of numerical invertibility, as also explained theoretically by Cornish et al. [11]. This is not too surprising, as attempting to learn a smooth invertible function mapping $\mathbb{R}^D$ to some low-dimensional manifold is an intrinsically ill-posed problem. This body of work strongly motivates the development of models whose support has matching topology – including dimension – to that of the true data distribution.

**Manifold flows**   A challenge to overcome for obtaining NFs on manifolds is the Jacobian-transpose-Jacobian determinant computation. Current approaches for NFs on manifolds approach this challenge in one of two ways. The first assumes the manifold is known beforehand [18, 53, 40], severely limiting applicability to low-dimensional data where the true manifold can realistically be known. The second group circumvents the computation of the Jacobian-transpose-Jacobian entirely through various heuristics. Kumar et al. [35] use a potentially loose lower bound of the log-likelihood, and do not explicitly enforce injectivity, resulting in a method for which the change-of-variables almost surely does not hold. Cunningham et al. [12] propose to convolve the manifold distribution with Gaussian noise, which results in the model having high-dimensional support. Finally, Brehmer and Cranmer [8] propose the method we described in subsection 2.2, where manifold learning and density estimation are done separately in order to avoid the log determinant computation. Concurrently to our work, Ross and Cresswell [54] proposed a rectangular flow construction which sacrifices some expressiveness but allows for exact likelihood evaluation.

**Why optimize the volume-change term?**   Learning $f_\theta$ and $h_\eta$ separately without the Jacobian of $f_\theta$ is concerning: even if $f_\theta$ maps to the correct manifold, it might unnecessarily expand and contract volume in such a way that makes correctly learning $h_\eta$ much more difficult than it needs to be. Looking ahead to our experiments, Figure 1 exemplifies this issue: the **top-middle** panel shows the ground truth density on a 1-dimensional circle in $\mathbb{R}^2$, and the **top-right** panel shows the distribution recovered by the two-step method of Brehmer and Cranmer [8]. We can see that, while the manifold is correctly recovered, the distribution on it is not. The **bottom-right** panel shows the speed at which $f_{\theta^*}$ maps $\mathbb{R}$ to $\mathcal{M}_{\theta^*}$: the top of the circle, which should have large densities, also has high speeds. Indeed, there is nothing in the objective discouraging $f_\theta$ to learn this behaviour, which implies that the corresponding low-dimensional distribution must be concentrated in a small region and thus making it harder to learn. The **bottom-middle** panel confirms this explanation: the learned low-dimensional distribution (dark red) does not match what it should (i.e. the distribution of $\{f_{\theta^*}^\dagger(x_i)\}_{i=1}^n$, in light red). This failure could have been avoided by learning the manifold in a density-aware fashion by including the Jacobian-transpose-Jacobian determinant in the objective.

## 4   Maximum Likelihood for Rectangular Flows: Taming the Gradient

### 4.1   Our Optimization Objective

We have argued that including the Jacobian-transpose-Jacobian in the optimization objective is sensible. However, as we previously mentioned, (5) corresponds to the density of the projection of $x$ onto $\mathcal{M}_\theta$. Thus, simply optimizing the likelihood would not result in learning $\mathcal{M}_\theta$ in such a way that observed data lies on it, only encouraging *projected* data points to have high likelihood. We thus maximize the log-likelihood subject to the constraint that the reconstruction error should be smaller than some threshold, i.e. $\phi^* = \arg\max_\phi \sum_{i=1}^n \log p_X(x_i)$ subject to $\sum_{i=1}^n ||x_i - f_\theta(f_\theta^\dagger(x_i))||_2^2 \leq \kappa$.

In practice, we use the KKT conditions [26, 34] and maximize the Lagrangian [7] instead:

$$\phi^* = \arg\max_\phi \sum_{i=1}^n \left\{ \log p_Z\left(g_\phi^\dagger(x_i)\right) - \log\left|\det \mathbf{J}[h_\eta]\left(g_\phi^\dagger(x_i)\right)\right| - \frac{1}{2}\log\det J_\theta^\top(x_i)J_\theta(x_i) \right. \quad (8)$$

$$\left. -\beta\left\|x_i - f_\theta\left(f_\theta^\dagger(x_i)\right)\right\|_2^2 \right\},$$

where we treat $\beta > 0$ as a hyperparameter rather than $\kappa$, and denote $\mathbf{J}[f_\theta](f_\theta^\dagger(x_i))$ as $J_\theta(x_i)$ for simplicity. We have dropped the absolute value since $J_\theta^\top(x_i)J_\theta(x_i)$ is always symmetric positive definite, since $J_\theta(x_i)$ has full rank by injectivity of $f_\theta$. We now make a technical but relevant observation about our objective: since our likelihoods are Radon-Nikodym derivatives with respect to a Riemannian measure on $\mathcal{M}_\theta$, different values of $\theta$ will result in different supports and dominating measures. One should thus be careful to compare likelihoods for models with different values of $\theta$. However, thanks to the smoothness of the objective over $\theta$, we should expect likelihoods for values of $\theta$ which are "close enough" to be comparable for practical purposes. In other words, comparisons remain reasonable locally, and the gradient of the volume-change term should contain relevant information to learn $\mathcal{M}_\theta$ in such a way that also facilitates learning $h_\eta$ on the pulled-back dataset $\{f_\theta^\dagger(x_i)\}_{i=1}^n$.

## 4.2 Optimizing our Objective: Stochastic Gradients

Note that all the terms in (8) are straightforward to evaluate and backpropagate through except for the third one; in this section we show how to obtain unbiased stochastic estimates of its gradient. In what follows we drop the dependence of the Jacobian on $x_i$ from our notation and write $J_\theta$, with the understanding that the end computation will be parallelized over a batch of $x_i$s. We assume access to an efficient matrix-vector product routine, i.e. computing $J_\theta^\top J_\theta \epsilon$ can be quickly achieved for any $\epsilon \in \mathbb{R}^d$. We elaborate on how we obtain these matrix-vector products in the next section. It is a well known fact from matrix calculus [49] that:

$$\frac{\partial}{\partial\theta_j}\log\det J_\theta^\top J_\theta = \text{tr}\left((J_\theta^\top J_\theta)^{-1}\frac{\partial}{\partial\theta_j}J_\theta^\top J_\theta\right), \quad (9)$$

where $\text{tr}$ denotes the trace operator and $\theta_j$ is the $j$-th element of $\theta$. Next, we can use Hutchinson's trace estimator [23], which states that for any matrix $M \in \mathbb{R}^{d\times d}$, $\text{tr}(M) = \mathbb{E}_\epsilon[\epsilon^\top M\epsilon]$ for any $\mathbb{R}^d$-valued random variable $\epsilon$ with zero mean and identity covariance matrix. We can thus obtain an unbiased stochastic estimate of our gradient as:

$$\frac{\partial}{\partial\theta_j}\log\det J_\theta^\top J_\theta \approx \frac{1}{K}\sum_{k=1}^K \epsilon_k^\top (J_\theta^\top J_\theta)^{-1}\frac{\partial}{\partial\theta_j}J_\theta^\top J_\theta\epsilon_k, \quad (10)$$

where $\epsilon_1, \ldots, \epsilon_K$ are typically sampled either from standard Gaussian or Rademacher distributions. Naïve computation of the above estimate remains intractable without explicitly constructing $J_\theta^\top J_\theta$. Fortunately, the $J_\theta^\top J_\theta\epsilon$ terms can be trivially obtained using the given matrix-vector product routine, avoiding the construction of $J_\theta^\top J_\theta$, and then $\partial/\partial\theta_j J_\theta^\top J_\theta\epsilon$ follows by taking the gradient w.r.t. $\theta$.

There is however still the issue of computing $\epsilon^\top(J_\theta^\top J_\theta)^{-1} = [(J_\theta^\top J_\theta)^{-1}\epsilon]^\top$. We use conjugate gradients (CG) [43] in order to achieve this. CG is an iterative method to solve problems of the form $Au = \epsilon$ for given $A \in \mathbb{R}^{d\times d}$ (in our case $A = J_\theta^\top J_\theta$) and $\epsilon \in \mathbb{R}^d$; we include the CG algorithm in Appendix B for completeness. CG has several important properties. First, it is known to recover the solution (assuming exact arithmetic) after at most $d$ steps, which means we can evaluate $A^{-1}\epsilon$. The solution converges exponentially (in the number of iterations $\tau$) to the true value [55], so often $\tau \ll d$ iterations are sufficient for accuracy to many decimal places. In practice, if we can tolerate a certain amount of bias, we can further increase computational speed by stopping iterations early. Second, CG only requires a method to compute matrix-vector products against $A$, and does not require access to $A$ itself. One such product is performed at each iteration, and CG thus requires at most $d$ matrix-vector products, though again in practice $\tau \ll d$ products usually suffice. This results in $\mathcal{O}(\tau d^2)$ solve complexity—less than the $\mathcal{O}(d^3)$ required by direct inversion methods. We denote $A^{-1}\epsilon$ computed with conjugate gradients as $\texttt{CG}(A; \epsilon)$. We can then compute the estimator from (10) as:

$$\frac{\partial}{\partial\theta_j}\log\det J_\theta^\top J_\theta \approx \frac{1}{K}\sum_{k=1}^K \texttt{CG}\left(J_\theta^\top J_\theta; \epsilon_k\right)^\top \frac{\partial}{\partial\theta_j}J_\theta^\top J_\theta\epsilon_k. \quad (11)$$

In practice, we implement this term by noting that $\text{CG}(J_\theta^\top J_\theta; \epsilon)^\top \partial/\partial\theta_j J_\theta^\top J_\theta\epsilon = \partial/\partial\theta_j\texttt{stop\_gradient}(\text{CG}(J_\theta^\top J_\theta; \epsilon)^\top)J_\theta^\top J_\theta\epsilon$, thereby taking advantage of the $\texttt{stop\_gradient}$ operation from Automatic Differentiation (AD) libraries and allowing us to avoid implementing a custom backward pass. We thus compute the contribution of a point $x$ to the training objective as:

$$\log p_Z\left(g_\phi^\dagger(x)\right) - \log\left|\det \mathbf{J}[h_\eta]\left(g_\phi^\dagger(x)\right)\right| - \beta\left\|\left|x - f_\theta\left(f_\theta^\dagger(x)\right)\right\|\right\|_2^2 \tag{12}$$

$$- \frac{1}{2K}\sum_{k=1}^K \texttt{stop\_gradient}\left(\text{CG}\left(J_\theta^\top J_\theta; \epsilon_k\right)^\top\right)J_\theta^\top J_\theta\epsilon_k$$

which gives the correct gradient estimate when taking the derivative with respect to $\phi$.

**Linear solvers for Jacobian terms** We note that linear solvers like CG have been used before to backpropagate through log determinant computations in the context of Gaussian processes [17], and more recently for square NFs with flexible architectures which do not allow for straightforward Jacobian determinant computations [22, 39]. However, none of these methods require the Jacobian-transpose-Jacobian-vector product that we derive in the next section, and to the best of our knowledge, these techniques have not been previously applied for training rectangular NFs. We also point out that recently Oktay et al. [44] proposed a method to efficiently obtain stochastic estimates of $J_\theta\epsilon$. While their method cannot be used as a drop-in replacement within our framework as it would result in a biased CG output, we believe this could be an interesting direction for future work. Finally, we note that CG has recently been combined with the Russian roulette estimator [25] to avoid having to always iterate $d$ times while maintaining unbiasedness, again in the context of Gaussian processes [50]. We also leave the exploration of this estimator within our method for future work.

## 4.3 AD Considerations: The Exact Method and the Forward-Backward AD Trick

In this section we derive the aforementioned routine for vector products against $J_\theta^\top J_\theta$, as well as an exact method that avoids the need for stochastic gradients (for a given $x$) at the price of increased memory requirements. But first, let us ask: why are these methods needed in the first place? There is work using power series to obtain stochastic estimates of log determinants [20, 9], and one might consider using them in our setting. However, these series require knowledge of the singular values of $J_\theta^\top J_\theta$, to which we do not have access (constructing $J_\theta^\top J_\theta$ to obtain its singular values would defeat the purpose of using the power series in the first place), and we would thus not have a guarantee that the series are valid. Additionally, they have to be truncated and thus result in biased estimators, and using Russian roulette estimators to avoid bias [9] can result in infinite variance [11]. Finally, these series compute and backpropagate (w.r.t. $\theta$) through products of the form $\epsilon^\top(J_\theta^\top J_\theta)^m\epsilon$ for different values of $m$, which can easily require more matrix-vector products than our methods. Behrmann et al. [4] address some of the issues with power series approximations as the result of controlling Lipschitz constants, although their estimates remain biased and potentially expensive.

Having motivated our approach, we now use commonly-known properties of AD to derive it; we briefly review these properties in Appendix C, referring the reader to Baydin et al. [3] for more detail. First, we consider the problem of explicitly constructing $J_\theta$. This construction can then be used to evaluate $J_\theta^\top J_\theta$ and exactly compute its log determinant either for log density evaluation of a trained model, or to backpropagate (with respect to $\theta$) through both the log determinant computation and the matrix construction, thus avoiding having to use stochastic gradients as in the previous section. We refer to this procedure as the *exact* method. Naïvely, one might try to explicitly construct $J_\theta$ using only backward-mode AD, which would require $D$ vector-Jacobian products (vjps) of the form $v^\top J_\theta$ – one per basis vector $v \in \mathbb{R}^D$ (and then stacking the resulting row vectors vertically). A better way to explicitly construct $J_\theta$ is with forward-mode AD, which only requires $d$ Jacobian-vector products (jvps) $J_\theta\epsilon$, again one per basis vector $\epsilon \in \mathbb{R}^d$ (and then stacking the resulting column vectors horizontally). We use a custom implementation of forward-mode AD in the popular PyTorch [47] library[2] for the exact method, as well as for the forward-backward AD trick described below.

We now explain how to combine forward- and backward-mode AD to obtain efficient matrix-vector products against $J_\theta^\top J_\theta$ in order to obtain the tractable gradient estimates from the previous section.

---

[2]PyTorch has a forward-mode AD implementation which relies on the "double backward" trick, which is known to be memory-inefficient. See https://j-towns.github.io/2017/06/12/A-new-trick.html for a description.

Table 1: Number of `jvps` and `vjps` (with respect to inputs) needed for forward and backward passes (with respect to $\theta$), along with the corresponding variance of gradient entries.

| Method | FORWARD | BACKWARD | VARIANCE |
|---|---|---|---|
| Exact (naïve) | $D$ `vjps` | $D$ `vjps` | $0$ |
| Exact | $d$ `jvps` | $d$ `jvps` | $0$ |
| Stochastic | $K(\tau+1)$`jvps` $+K(\tau+1)$ `vjps` | $K$`jvps` $+K$`vjps` | $\propto 1/K$ |

Note that $v := J_\theta \epsilon$ can be computed with a single `jvp` call, and then $J_\theta^\top J_\theta \epsilon = [v^\top J_\theta]^\top$ can be efficiently computed using only a `vjp` call. We refer to this way of computing matrix-vector products against $J_\theta^\top J_\theta$ as the *forward-backward AD trick*. We summarize both of our gradient estimators in Appendix D. Note that (12) requires $K(\tau+1)$ such matrix-vector products, which is seemingly less efficient as it is potentially greater than the $d$ `jvps` required by the exact method. However, the stochastic method is much more memory-efficient than its exact counterpart when optimizing over $\theta$: of the $K(\tau+1)$ matrix-vector products needed to evaluate (12), only $K$ require gradients with respect to $\theta$. Thus only $K$ `jvps` and $K$ `vjps`, along with their intermediate steps, must be stored in memory over a training step. In contrast, the exact method requires gradients (w.r.t. $\theta$) for every one of its $d$ `jvp` computations, which requires storing these computations along with their intermediate steps in memory.

Our proposed methods thus offer a memory vs. variance trade-off. Increasing $K$ in the stochastic method results in larger memory requirements which imply longer training times, as the batch size must be set to a smaller value. On the other hand, the larger the memory cost, the smaller the variance of the gradient. This still holds true for the exact method, which results in exact gradients, at the cost of increased memory requirements (as long as $K \ll d$; if $K$ is large enough the stochastic method should never be used over the exact one). Table 1 summarizes this trade-off.

## 5 Experiments

We now compare our methods against the two-step baseline of Brehmer and Cranmer [8], and also study the memory vs. variance trade-off. We use the real NVP [15] architecture for all flows, except we do not use batch normalization [24] as it causes issues with `vjp` computations. We point out that all comparisons remain fair, and we include a detailed explanation of this phenomenon in Appendix E, along with all experimental details in Appendix G. Throughout, we use the abbreviations RNFs-ML for our maximum likelihood training method, RNFs-TS for the two-step method, and RNFs for rectangular NFs in general. For most runs, we found it useful to anneal the likelihood term(s). That is, at the beginning of training we optimize only the reconstruction term, and then slowly incorporate the other terms. This likelihood annealing procedure helped avoid local optima where the manifold is not recovered (large reconstruction error) but the likelihood of projected data is high.

### 5.1 Simulated Data

We consider a simulated dataset where we have access to ground truth, which allows us to empirically verify the deficiencies of RNFs-TS. We use a von Mises distribution, which is supported on the one-dimensional unit circle in $\mathbb{R}^2$. Figure 1 shows this distribution, along with its estimates from RNFs-ML (exact) and RNFs-TS. As previously observed, RNFs-TS correctly approximate the manifold, but fail to learn the right distribution on it. In contrast we can see that RNFs-ML, by virtue of including the Jacobian-transpose-Jacobian term in the optimization, manage to recover both the manifold and the distribution on it (**top left panel**), while also resulting in an easier-to-learn low-dimensional distribution (**bottom middle panel**) thanks to $f_{\theta^*}$ mapping to $\mathcal{M}_{\theta^*}$ at a more consistent speed (**bottom left panel**). We do point out that, while the results presented here are representative of usual runs for both methods, we did have runs with different results which we include in Appendix G for completeness. We finish with the observation that even though the line and the circle are not homeomorphic and thus RNFs are not perfectly able to recover the support, they manage to adequately approximate it.

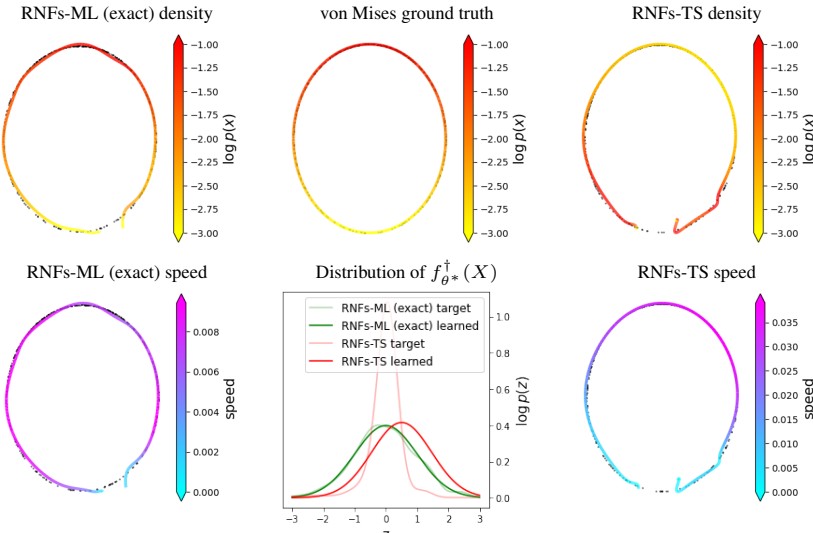

Figure 1: Top row: RNFs-ML (exact) (left), von Mises ground truth (middle), and RNF-TS (right). Bottom row: Speed at which $f_{\theta*}$ maps to $\mathcal{M}_{\theta*}$ (measured as $l_2$ distance between uniformly spaced consecutive points in $\mathbb{R}$ mapped through $f_{\theta*}$) for RNFs-ML (exact) (left), RNFs-TS (right), and distribution $h_\eta$ has to learn in order to recover the ground truth, fixing $\theta^*$ (middle). See text for discussion.

Table 2: FID-like metric for tabular data (lower is better). Bolded runs are the best or overlap with it.

| Method | POWER | GAS | HEPMASS | MINIBOONE |
|---|---|---|---|---|
| RNFs-ML (exact) | $\mathbf{0.067 \pm 0.016}$ | $\mathbf{0.138 \pm 0.023}$ | $\mathbf{0.486 \pm 0.032}$ | $\mathbf{0.978 \pm 0.082}$ |
| RNFs-ML ($K = 1$) | $\mathbf{0.083 \pm 0.015}$ | $\mathbf{0.110 \pm 0.021}$ | $0.779 \pm 0.191$ | $\mathbf{1.001 \pm 0.051}$ |
| RNFs-ML ($K = 10$) | $0.113 \pm 0.037$ | $\mathbf{0.140 \pm 0.013}$ | $\mathbf{0.495 \pm 0.055}$ | $\mathbf{0.878 \pm 0.083}$ |
| RNFs-TS | $0.178 \pm 0.024$ | $0.161 \pm 0.016$ | $0.649 \pm 0.081$ | $1.085 \pm 0.062$ |

## 5.2 Tabular Data

We now turn our attention to the tabular datasets used by Papamakarios et al. [45], now a common benchmark for NFs as well. As previously mentioned, one should be careful when comparing models with different supports, as we cannot rely on test likelihoods as a metric. We take inspiration from the FID score [21], which is commonly used to evaluate quality of generated images when likelihoods are not available. The FID score compares the first and second moments of a well-chosen statistic – taken in practice to be the values of the last hidden layer of a pre-trained inception network [56] – from the model and data distributions using the squared Wasserstein-2 metric (between Gaussians). Here, we take the statistic to be the data itself instead of the final hidden units of a pre-trained classifier: in other words, our metric compares the mean and covariance of generated data against those of observed data with the same squared Wasserstein-2 metric. We include the mathematical formulas for computing both FID and our modified version for tabular data in Appendix F. We use early stopping with our FID-like score across all models. Our results are summarized in Table 2, where we can see that RNFs-ML consistently do a better job at recovering the underlying distribution. Once again, these results emphasize the benefits of including the Jacobian-transpose-Jacobian in the objective. Interestingly, except for HEPMASS, the results from our stochastic version with $K = 1$ are not significantly exceeded by the exact version or using a larger value of $K$, suggesting that the added variance does not result in decreased empirical performance. We highlight that no tuning was done (except on GAS for which we changed $d$ from 4 to 2), RNFs-ML outperformed RNFs-TS out-of-the-box here (details are in Appendix G). We report training times in Appendix G, and observe that RNFs-ML take a similar amount of time as RNFs-TS to train for datasets with lower values of $D$, and while we do take longer to train for the other datasets, our training times remain reasonable and we often require fewer epochs to converge.

Table 3: FID scores (lower is better) and decision stump OoD accuracy (higher is better).

| Method | FID | | | OoD ACCURACY | |
|---|---|---|---|---|---|
| | CIFAR-10 | MNIST | FMNIST | MNIST $\to$ FMNIST | FMNIST $\to$ MNIST |
| RNFs-ML (exact) | **643.31** | 36.09 | 296.01 | 92% | 91% |
| RNFs-ML ($K = 1$) | 830.94 | **33.98** | **288.39** | 97% | 78% |
| RNFs-ML ($K = 4$) | - | 42.90 | 342.91 | 77% | 89% |
| RNFs-TS | 731.46 | 35.52 | 318.59 | **98%** | **96%** |

## 5.3 Image Data and Out-of-Distribution Detection

We also compare RNFs-ML to RNFs-TS for image modelling on MNIST and FMNIST. We point out that these datasets have ambient dimension $D = 784$, and being able to fit RNFs-ML is in itself noteworthy: to the best of our knowledge no previous method has scaled optimizing the Jacobian-transpose-Jacobian term to these dimensions. We use FID scores both for comparing models and for early stopping during training. We also used likelihood annealing, with all experimental details again given in Appendix G. We report FID scores in Table 3, where we can see that we outperform RNFs-TS. Our RNFs-ML ($K = 1$) variant also outperforms its decreased-variance counterparts. This initially puzzling behaviour is partially explained by the fact that we used the $K = 1$ variant to tune the model (being the cheapest one to train), and then used the tuned hyperparameters for a single run of the other two variants. Nonetheless, once again these results suggest that the variance induced by our stochastic method is not empirically harmful, and that while using the exact method should be the default whenever feasible, using $K = 1$ otherwise is sensible. We also report training times where we can see the computational benefits of our stochastic method, as well as visualizations of samples, in Appendix G.

We also compare performance on the CIFAR-10 dataset [33], for which $D = 3,072$. Once again, being able to fit RNFs-ML in this setting is in itself remarkable. We do not include RNFs-ML ($K = 4$) results because of limited experimentation on CIFAR-10 due to computational cost (experimental details, including hyperparameters which we tried, are given in Appendix G). We can see that, while RNFs-TS outperformed RNFs-ML ($K = 1$) – which we hypothesize might be reversed given more tuning – our RNFs-ML (exact) version is the best performing model, yet again highlighting the importance of including the change-of-volume term in the objective.

We further evaluate the performance of RNFs for OoD detection. Nalisnick et al. [41] pointed out that square NFs trained on FMNIST assign higher likelihoods to MNIST than they do to FMNIST. While there has been research attempting to fix this puzzling behaviour [1, 2, 10, 51], to the best of our knowledge no method has managed to correct it using only likelihoods of trained models. Figure 2 shows that RNFs remedy this phenomenon, and that models trained on FMNIST assign higher test likelihoods to FMNIST than to MNIST. This correction does not come at the cost of strange behaviour now emerging in the opposite direction (i.e. when training on MNIST, see Appendix G for a histogram). Table 3 quantifies these results (arrows point from in-distribution datasets to OoD ones) with the accuracy of a decision stump using only log-likelihood, and we can see that the best-performing RNFs models essentially solve this OoD task. While we leave a formal explanation of this result for future work, we believe this discovery highlights the importance of

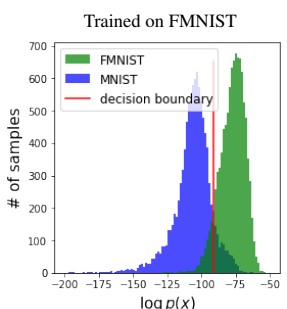

Figure 2: OoD detection with RNFs-ML (exact).

properly specifying models and of ensuring the use of appropriate inductive biases, in this case low intrinsic dimensionality of the observed data. The strong performance of RNFs-TS here seems to indicate that this is a property of RNFs rather than of our ML training method specifically, although our exact approach is still used to compute these log-likelihoods at test time. We include additional results on OoD detection using reconstruction errors – along with a discussion – in Appendix G, where we found the opposite unexpected behaviour: FMNIST always has smaller reconstruction errors, regardless of which dataset was used for training.

# 6    Scope and Limitations

In this paper we address the dimensionality-based misspecification of square NFs while properly using maximum likelihood as the training objective, thus providing an advancement in the training of RNFs. Our methods however remain *topologically* misspecified: even though we can better address dimensionality, we can currently only learn manifolds homeomorphic to $\mathbb{R}^d$. For example, one could conceive of the MNIST manifold as consisting of 10 connected components (one per digit), which cannot be learned by $f_\theta$. It is nonetheless worth noting that this limitation is shared by other deep generative modelling approaches, for example GANs [19] result in connected supports (since the image of a connected set under a continuous function is connected). We observed during training in image data that the residuals of CG were not close to $\mathbf{0}$ numerically, even after $d$ steps, indicating poor conditioning and thus possible numerical non-invertibility of the matrix $J_\theta^\top J_\theta$. We hypothesize that this phenomenon is caused by topological mismatch, which we also conjecture affects us more than the baseline as our CG-obtained (or from the exact method) gradients might point in an inaccurate direction. We thus expect our methods in particular to benefit from improved research on making flows match the target topology, for example via continuous indexing [11].

Additionally, while we have successfully scaled likelihood-based training of RNFs far beyond current capabilities, our methods – even the stochastic one – remain computationally expensive for higher dimensions, and further computational gains remain an open problem. We also attempted OoD detection on CIFAR-10 against the SVHN dataset [42], and found that neither RNFs-ML nor RNFs-TS has good performance, although anecdotally we may have at least improved on the situation outlined by Nalisnick et al. [41]. We hypothesize these results might be either caused by topological mismatch, or corrected given more tuning.

# 7    Conclusions and Broader Impact

In this paper we argue for the importance of likelihood-based training of rectangular flows, and introduce two methods allowing to do so. We study the benefits of our methods, and empirically show that they are preferable to current alternatives. Given the methodological nature of our contributions, we do not foresee our work having any negative ethical implications or societal consequences.

## Acknowledgements

We thank Brendan Ross, Jesse Cresswell, and Maksims Volkovs for useful comments and feedback. We would also like to thank Rob Cornish for the excellent CIFs codebase upon which our code is built, and Emile Mathieu for plotting suggestions. GP and JPC are supported by the Simons Foundation, McKnight Foundation, the Grossman Center, and the Gatsby Charitable Trust.

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
