## A Injective Change-of-Variable Formula and Stacking Injective Flows

We first derive (5) from (3). By the chain rule, we have:

$$\mathbf{J}[g_\phi]\left(g_\phi^\dagger(x)\right) = \mathbf{J}[f_\theta]\left(f_\theta^\dagger(x)\right)\mathbf{J}[h_\eta]\left(g_\phi^\dagger(x)\right). \tag{13}$$

The Jacobian-transpose Jacobian term in (3) thus becomes:

$$\left|\det \mathbf{J}[g_\phi]^\top\left(g_\phi^\dagger(x)\right)\mathbf{J}[g_\phi]\left(g_\phi^\dagger(x)\right)\right|^{-1/2} \tag{14}$$

$$= \left|\det \mathbf{J}[h_\eta]^\top\left(g_\phi^\dagger(x)\right)\mathbf{J}[f_\theta]^\top\left(f_\theta^\dagger(x)\right)\mathbf{J}[f_\theta]\left(f_\theta^\dagger(x)\right)\mathbf{J}[h_\eta]\left(g_\phi^\dagger(x)\right)\right|^{-1/2}$$

$$= \left|\det \mathbf{J}[h_\eta]^\top\left(g_\phi^\dagger(x)\right)\right|^{-1/2}\left|\det \mathbf{J}[f_\theta]^\top\left(f_\theta^\dagger(x)\right)\mathbf{J}[f_\theta]\left(f_\theta^\dagger(x)\right)\right|^{-1/2}\left|\det \mathbf{J}[h_\eta]\left(g_\phi^\dagger(x)\right)\right|^{-1/2}$$

$$= \left|\det \mathbf{J}[h_\eta]\left(g_\phi^\dagger(x)\right)\right|^{-1}\left|\det \mathbf{J}[f_\theta]^\top\left(f_\theta^\dagger(x)\right)\mathbf{J}[f_\theta]\left(f_\theta^\dagger(x)\right)\right|^{-1/2},$$

where the second equality follows from the fact that $\mathbf{J}[h_\eta]^\top(g_\phi^\dagger(x))$, $\mathbf{J}[f_\theta]^\top(f_\theta^\dagger(x))\mathbf{J}[f_\theta](f_\theta^\dagger(x))$, and $\mathbf{J}[h_\eta](g_\phi^\dagger(x))$ are all square $d \times d$ matrices; and the third equality follows because determinants are invariant to transpositions. The observation that the three involved matrices are square is the reason behind why we can decompose the change-of-variable formula for $g_\phi$ as applying first the change-of-variable formula for $h_\eta$, and then applying it for $f_\theta$.

This property, unlike in the case of square flows, does not always hold. That is, the change-of-variable formula for a composition of injective transformations is not necessarily equivalent to applying the injective change-of-variable formula twice. To see this, consider the case where $g_1 : \mathbb{R}^d \to \mathbb{R}^{d_2}$ and $g_2 : \mathbb{R}^{d_2} \to \mathbb{R}^D$ are injective, where $d < d_2 < D$ and let $g = g_2 \circ g_1$. Clearly $g$ is injective by construction, and thus the determinant from its change-of-variable formula at a point $z \in \mathbb{R}^d$ is given by:

$$\det \mathbf{J}[g]^\top(z)\mathbf{J}[g](z) = \det \mathbf{J}[g_1]^\top(z)\mathbf{J}[g_2]^\top\left(g_1(z)\right)\mathbf{J}[g_2]\left(g_1(z)\right)\mathbf{J}[g_1](z), \tag{15}$$

where now $\mathbf{J}[g_1](z) \in \mathbb{R}^{d_2 \times d}$ and $\mathbf{J}[g_2](g_1(z)) \in \mathbb{R}^{D \times d_2}$. Unlike the determinant from (14), this determinant cannot be easily decomposed into a product of determinants since the involved matrices are not all square. In particular, (15) need not match:

$$\det \mathbf{J}[g_1]^\top(z)\mathbf{J}[g_1](z) \cdot \det \mathbf{J}[g_2]^\top(g_1(z))\mathbf{J}[g_2](g_1(z)), \tag{16}$$

which would be the determinant terms from applying the change-of-variable formula twice. Note that this observation does not imply that a flow like $g$ could not be trained with our method, it simply implies that the $\det \mathbf{J}[g]^\top(z)\mathbf{J}[g](z)$ term has to be considered as a whole, and not decomposed into separate terms. It is easy to verify that in general, only an initial $d$-dimensional square flow can be separated from the overall Jacobian-transpose-Jacobian determinant.

## B Conjugate Gradients

We outline the CG algorithm in Algorithm 1, whose output we write as $\texttt{CG}(A; \epsilon)$ in the main manuscript (we omit the dependance on the tolerance $\delta$ for notational simplicity). Note that CG does not need access to $A$, just a matrix-vector product routine against $A$, $\texttt{mvp\_A}(\cdot)$. If $A$ is symmetric positive definite, then CG converges in at most $d$ steps, i.e. its output matches $A^{-1}\epsilon$ and the corresponding residual is 0, and CG uses thus at most $d$ calls to $\texttt{mvp\_A}(\cdot)$. This convergence

holds mathematically, but can be violated numerically if $A$ is ill-conditioned, which is why the $\tau < d$ condition is added in the while loop.

---

**Algorithm 1:** CG

---

**Input** : $\mathtt{mvp\_A}(\cdot)$, function for matrix-vector products against $A \in \mathbb{R}^{d \times d}$
        $\epsilon \in \mathbb{R}^d$
        $\delta \geq 0$, tolerance
**Output** : $A^{-1}\epsilon$
$u_0 \leftarrow \mathbf{0} \in \mathbb{R}^d$ // current solution
$r_0 \leftarrow -\epsilon$ // current residual
$q_0 \leftarrow r_0$
$\tau \leftarrow 0$
**while** $\|r_\tau\|_2 > \delta$ **and** $\tau < d$ **do**
    $v_\tau \leftarrow \mathtt{mvp\_A}(q_\tau)$
    $\alpha_\tau \leftarrow (r_\tau^\top r_\tau)/(q_\tau^\top v_\tau)$
    $u_{\tau+1} \leftarrow u_\tau + \alpha_\tau q_\tau$
    $r_{\tau+1} \leftarrow r_\tau - \alpha_\tau v_\tau$
    $\beta_\tau \leftarrow (r_{\tau+1}^\top r_{\tau+1})/(r_\tau^\top r_\tau)$
    $q_{\tau+1} \leftarrow r_{\tau+1} + \beta_\tau q_\tau$
    $\tau \leftarrow \tau + 1$
**end**
**return** $u_\tau$

---

## C  Automatic Differentiation

Here we summarize the relevant properties from forward- and backward-mode automatic differentiation (AD) which we use in the main manuscript. Let $f$ be the composition of smooth functions $f_1, \ldots, f_L$, i.e. $f = f_L \circ f_{L-1} \circ \cdots \circ f_1$. For example, in our setting this function could be $f_\theta$, so that $f_1 = \mathtt{pad}$, and the rest of the functions could be coupling layers from a $D$-dimensional square flow (or the functions whose compositions results in the coupling layers). By the chain rule, the Jacobian of $f$ is given by:

$$\mathbf{J}[f](z) = \mathbf{J}[f_L](\bar{f}_{L-1}(z)) \cdots \mathbf{J}[f_2](\bar{f}_1(z))\mathbf{J}[f_1](z), \tag{17}$$

where $\bar{f}_l := f_l \circ f_{l-1} \circ \cdots \circ f_1$ for $l = 1, 2, \ldots, L-1$. Forward-mode AD computes products from right to left, and is thus efficient for computing $\mathtt{jvp}$ operations. Computing $\mathbf{J}[f](z)\epsilon$ is thus obtained by performing $L$ matrix-vector multiplications, one against each of the Jacobians on the right hand side of (17). Backward-mode AD computes products from left to right, and would thus result in significantly more inefficient $\mathtt{jvp}$ evaluations involving $L-1$ matrix-matrix products, and a single matrix-vector product. Analogously, backward-mode AD computes $\mathtt{vjps}$ of the form $v^\top \mathbf{J}[f](z)$ efficiently, using $L$ vector-matrix products, while forward-mode AD would require $L-1$ matrix-matrix products and a single vector-matrix product.

Typically, the cost of evaluating a matrix-vector or vector-matrix product against $\mathbf{J}[f_{l+1}](\bar{f}_l)$ (or $\mathbf{J}[f_1](z)$) is the same as computing $\bar{f}_{l+1}(z)$ from $\bar{f}_l(z)$, i.e. the cost of evaluating $f_{l+1}$ (or the cost of evaluating $f_1$ in the case of $\mathbf{J}[f_1](z)$) [3]. $\mathtt{jvp}$ and $\mathtt{vjp}$ computations thus not only have the same computational cost, but this cost is also equivalent to a forward pass, i.e. computing $f$.

When computing $f$, obtaining a $\mathtt{jvp}$ with forward-mode AD adds the same memory cost as another computation of $f$ since intermediate results do not have to be stored. That is, in order to compute $\mathbf{J}[f_l](\bar{f}_{l-1}(z)) \cdots \mathbf{J}[f_1](z)\epsilon$, we only need to store $\mathbf{J}[f_{l-1}](\bar{f}_{l-2}(z)) \cdots \mathbf{J}[f_1](z)\epsilon$ and $\bar{f}_{l-1}(z)$ (which has to be stored anyway for computing $f$) in memory. On the other hand, computing a $\mathtt{vjp}$ with backward-mode AD has a higher memory cost: One has to first compute $f$ and store all the intermediate $\bar{f}_l(z)$ (along with $z$), since computing $v^\top \mathbf{J}[f_L](\bar{f}_{L-1}(z)) \cdots \mathbf{J}[f_l](\bar{f}_{l-1}(z))$ from $v^\top \mathbf{J}[f_L](\bar{f}_{L-1}(z)) \cdots \mathbf{J}[f_{l+1}](\bar{f}_l(z))$ requires having $\bar{f}_{l-1}(z)$ in memory.

In practice, we use PyTorch's implementation of backpropagation to compute $\mathtt{vjps}$, and as mentioned in the main manuscript, we use our own implementation of forward-mode AD for $\mathtt{jvps}$. We achieve this by having every layer and non-linearity $f_l$ in our networks not only take an input $x$, but also a vector $\epsilon$ of the same length as $x$; and not just output the usual output $f_l(x)$, but also $\mathbf{J}[f_l](x)\epsilon$

(for linear layers, this is equivalent to applying the layer without the bias to $\epsilon$, and for element-wise non-linearities $\mathbf{J}[f_l](x)$ is a straightforward-to-compute diagonal matrix and so $\mathbf{J}[f_l](x)\epsilon$ can be obtained though element-wise products).

# D   Summary of our Proposed Methods

We summarize our methods for computing/estimating the gradient of the log determinant arising in maximum likelihood training of rectangular flows. Algorithm 2 shows the exact method, where $\texttt{jvp}(f, z, \epsilon)$ denotes computing $\mathbf{J}[f](z)\epsilon$ using forward-mode AD, and $\epsilon_i \in \mathbb{R}^d$ is the $i$-th standard basis vector, i.e. a one-hot vector with a 1 on its $i$-th coordinate. Note that $\partial/\partial\theta \log \det A_\theta$ is computed using backpropagation. The $\texttt{for}$ loop is easily parallelized in practice. For density evaluation, rather than returning $\partial/\partial\theta \log \det A_\theta$, the output of Algorithm 2 becomes $\log \det A_\theta$.

Algorithm 3 shows our stochastic method, where $\texttt{vjp}(f, z, v)$ denotes $v^\top \mathbf{J}[f](z)$ computed through backward-mode AD. As mentioned in the main manuscript, the $\epsilon_k$ vectors can be sampled from any zero-mean, identity-covariance distribution and not just a Gaussian. For added clarity, we change the CG notation and use $\texttt{CG}(\texttt{mvp\_A}(\cdot), \epsilon, \delta)$ to denote the output of the conjugate gradients method. Backpropagation is once again used to compute $\partial/\partial\theta(s_\theta/K)$, and the $\texttt{for}$ loop is again parallelized. Note that, unlike Algorithm 2, $s_\theta/K$ is not a valid log determinant estimate, and Algorithm 3 should *only* be used for gradient estimates during training, and not density evaluation at test time.

---

**Algorithm 2:** Exact method

**Input**  : $f_\theta : \mathbb{R}^d \to \mathbb{R}^D$
    $x \in \mathbb{R}^D$
**Output** : $\partial/\partial\theta \log \det J_\theta^\top J_\theta$

$z \leftarrow f_\theta^\dagger(x)$
**for** $i = 1, \dots, d$ **do**
 |  $v_i \leftarrow \texttt{jvp}(f_\theta, z, \epsilon_i)$
**end**
$J_\theta \leftarrow (v_1 | \dots | v_d)$
$A_\theta \leftarrow J_\theta^\top J_\theta$
**return** $\partial/\partial\theta \log \det A_\theta$

---

**Algorithm 3:** Stochastic method

**Input**  : $f_\theta : \mathbb{R}^d \to \mathbb{R}^D$
    $x \in \mathbb{R}^D$
    $K \in \mathbb{N}_+$
    $\delta \geq 0$, CG tolerance
**Output** : Unbiased stochastic approximation of $\partial/\partial\theta \log \det J_\theta^\top J_\theta$

$z \leftarrow f_\theta^\dagger(x)$
$\texttt{mvp\_A}(\cdot) \leftarrow \texttt{vjp}(f_\theta, z, \texttt{jvp}(f_\theta, z, \cdot))^\top$
$s_\theta \leftarrow 0$
**for** $k = 1, \dots, K$ **do**
 |  $\epsilon_k \sim \mathcal{N}(0, I_d)$
 |  $s_\theta \leftarrow s_\theta + \texttt{stop\_gradient}(\texttt{CG}(\texttt{mvp\_A}(\cdot), \epsilon_k, \delta)^\top) \cdot \texttt{mvp\_A}(\epsilon_k)$
**end**
**return** $\partial/\partial\theta(s_\theta/K)$

---

# E   Batch Normalization

We now explain the issues that arise when combining batch normalization with $\texttt{vjps}$. These issues arise not only in our setting, but every time backward-mode AD has to be called to compute or approximate the gradient of the determinant term. We consider the case with a batch of size 2, $x_1$ and $x_2$, as it exemplifies the issue and the notation becomes simpler. Consider applying $f_\theta$ (without batch normalization) to each element in the batch, which we denote with the batch function $F_\theta$:

$$F_\theta(x_1, x_2) := (f_\theta(x_1), f_\theta(x_2)).\tag{18}$$

The Jacobian of $F_\theta$ clearly has a block-diagonal structure:

$$\mathbf{J}[F_\theta](x_1, x_2) = \begin{pmatrix} \mathbf{J}[f_\theta](x_1) & \mathbf{0} \\ \mathbf{0} & \mathbf{J}[f_\theta](x_2) \end{pmatrix}. \tag{19}$$

This structure implies that relevant computations such as `vjps`, `jvps`, and determinants parallelize over the batch:

$$(v_1, v_2)^\top \mathbf{J}[F_\theta](x_1, x_2) = \left(v_1^\top \mathbf{J}[f_\theta](x_1), v_2^\top \mathbf{J}[f_\theta](x_2)\right) \tag{20}$$

$$\mathbf{J}[F_\theta](x_1, x_2) \begin{pmatrix} \epsilon_1 \\ \epsilon_2 \end{pmatrix} = \begin{pmatrix} \mathbf{J}[f_\theta](x_1)\epsilon_1 \\ \mathbf{J}[f_\theta](x_2)\epsilon_2 \end{pmatrix}$$

$$\det \mathbf{J}[F_\theta]^\top(x_1, x_2)\mathbf{J}[F_\theta](x_1, x_2) = \det \mathbf{J}[f_\theta]^\top(x_1)\mathbf{J}[f_\theta](x_1) \det \mathbf{J}[f_\theta]^\top(x_2)\mathbf{J}[f_\theta](x_2).$$

In contrast, when using batch normalization, the resulting computation $F_\theta^{BN}(x_1, x_2)$ does not have a block-diagonal Jacobian, and thus this parallelism over the batch breaks down, in other words:

$$(v_1, v_2)^\top \mathbf{J}\left[F_\theta^{(BN)}\right](x_1, x_2) \neq \left(v_1^\top \mathbf{J}[f_\theta](x_1), v_2^\top \mathbf{J}[f_\theta](x_2)\right) \tag{21}$$

$$\mathbf{J}\left[F_\theta^{BN}\right](x_1, x_2) \begin{pmatrix} \epsilon_1 \\ \epsilon_2 \end{pmatrix} \neq \begin{pmatrix} \mathbf{J}[f_\theta](x_1)\epsilon_1 \\ \mathbf{J}[f_\theta](x_2)\epsilon_2 \end{pmatrix}$$

$$\det \mathbf{J}\left[F_\theta^{BN}\right]^\top(x_1, x_2)\mathbf{J}\left[F_\theta^{BN}\right](x_1, x_2) \neq \det \mathbf{J}[f_\theta]^\top(x_1)\mathbf{J}[f_\theta](x_1) \det \mathbf{J}[f_\theta]^\top(x_2)\mathbf{J}[f_\theta](x_2),$$

where the above $\neq$ signs should be interpreted as "not generally equal to" rather than always not equal to, as equalities could hold coincidentally in rare cases.

In square flow implementations, AD is never used to obtain any of these quantities, and the Jacobian log determinants are explicitly computed for each element in the batch. In other words, this batch dependence is ignored in square flows, both in the log determinant computation, and when back-propagating through it. Elaborating on this point, AD is only used to backpropagate (with respect to $\theta$) over this explicit computation. If AD was used on $F_\theta^{BN}$ to construct the matrices and we then computed the corresponding log determinants, the results would not match with the explicitly computed log determinants: The latter would be equivalent to using batch normalization with a `stop_gradient` operation *with respect to $(x_1, x_2)$ but not with respect to $\theta$*, while the former would use no `stop_gradient` whatsoever. Unfortunately, this partial `stop_gradient` operation only with respect to inputs but not parameters is not available in commonly used AD libraries. While our custom implementation of `jvps` can be easily "hard-coded" to have this behaviour, doing so for `vjps` would require significant modifications to PyTorch. We note that this is *not* a fundamental limitation and that these modifications could be done to obtain `vjps` that behave as expected with a low-level re-implementation of batch normalization, but these fall outside of the scope of our paper. Thus, in the interest of performing computations in a manner that remains consistent with what is commonly done for square flows and that allows fair comparisons of our exact and stochastic methods, we avoid using batch normalization.

## F   FID and FID-like Scores

For a given dataset $\{x_1, \ldots, x_n\} \subset \mathbb{R}^D$ and a set of samples generated by a model $\{x_1^{(g)}, \ldots, x_m^{(g)}\} \subset \mathbb{R}^D$, along with a statistic $T : \mathbb{R}^D \to \mathbb{R}^r$, the empirical means and covariances are given by:

$$\hat{\mu} := \frac{1}{n} \sum_{i=1}^{n} T(x_i), \qquad \hat{\Sigma} := \frac{1}{n-1} \sum_{i=1}^{n} (T(x_i) - \hat{\mu})(T(x_i) - \hat{\mu})^\top \tag{22}$$

$$\hat{\mu}^{(g)} := \frac{1}{m} \sum_{i=1}^{m} T\left(x_i^{(g)}\right), \quad \hat{\Sigma}^{(g)} := \frac{1}{m-1} \sum_{i=1}^{m} \left(T\left(x_i^{(g)}\right) - \hat{\mu}^{(g)}\right)\left(T\left(x_i^{(g)}\right) - \hat{\mu}^{(g)}\right)^\top. \tag{23}$$

The FID score takes $T$ as the last hidden layer of a pre-trained inception network, and evaluates generated sample quality by comparing generated moments against data moments. This comparison is done with the squared Wasserstein-2 distance between Gaussians with corresponding moments, which is given by:

$$\left\|\hat{\mu} - \hat{\mu}^{(g)}\right\|_2^2 + \mathrm{tr}\left(\hat{\Sigma} + \hat{\Sigma}^{(g)} - 2\left(\hat{\Sigma}\hat{\Sigma}^{(g)}\right)^{1/2}\right), \tag{24}$$

Table 4: Training times in seconds, "$K > 1$" means $K = 10$ for tabular data and $K = 4$ for images.

| Dataset | RNFs-ML (exact) | | RNFs-ML ($K = 1$) | | RNFs-ML ($K > 1$) | | RNFs-TS | |
|---|---|---|---|---|---|---|---|---|
| | EPOCH | TOTAL | EPOCH | TOTAL | EPOCH | TOTAL | EPOCH | TOTAL |
| POWER | 53.8 | 4.13e3 | 67.4 | 6.76e3 | 136 | 1.14e4 | 45.1 | 3.83e3 |
| GAS | 37.3 | 2.51e3 | 62.7 | 4.51e3 | 80.1 | 5.24e3 | 43.2 | 3.49e3 |
| HEPMASS | 143 | 1.01e4 | 146 | 8.28e3 | 159 | 1.20e4 | 29.1 | 2.42e3 |
| MINIBOONE | 49.3 | 4.16e3 | 26.3 | 2.01e3 | 29.8 | 2.94e3 | 4.61 | 481 |
| MNIST | 2.40e3 | 2.59e5 | 1.71e3 | 1.57e5 | 3.03e3 | 3.20e5 | 2.13$e$2 | 3.90e4 |
| FMNIST | 2.34e3 | 2.59e5 | 1.72e3 | 1.50e5 | 3.15e3 | 2.10e5 | 1.04e2 | 1.11e4 |

which is 0 if and only if the moments match. Our proposed FID-like score for tabular data is computed the exact same way, except no inception network is used. Instead, we simply take $T$ to be the identity, $T(x) = x$.

## G Experimental Details

First we will comment on hyperparameters/architectural choices shared across experiments. The $D$-dimensional square flow that we use, as mentioned in the main manuscript, is a RealNVP network [15]. In all cases, we use the ADAM [27] optimizer and train with early stopping against some validation criterion specified for each experiment separately and discussed further in each of the relevant subsections below. We use no weight decay. We also do not use batch normalization in any experiments for the reasons mentioned above in Appendix E. We use a standard Gaussian on $d$ dimensions as $p_Z$ in all experiments.

**Compute**  We ran our two-dimensional experiments on a Lenovo T530 laptop with an Intel i5 processor, with negligible training time per epoch. We ran the tabular data experiments on a variety of NVIDIA GeForce GTX GPUs on a shared cluster: we had, at varying times, access to 1080, 1080 Ti, and 2080 Ti models, but never access to more than six cards in total at once. For the image experiments, we had access to a 32GB-configuration NVIDIA Tesla v100 GPU. We ran each of the tabular and image experiments on a single card at a time, except for the image experiments for the RNFs-ML (exact) and ($K = 10$) models which we parallelized over four cards.

Table 4 includes training times for all of our experiments. Since we used FID-like and FID scores for ealy stopping, we include both per-epoch and total times. Per epoch times of RNFs-ML exclude epochs where the Jacobian-transpose-Jacobian log determinant is annealed with a 0 weight, although we include time added from this portion of training into the total time cost. Note throughout this section we also consider one epoch of the two-step baseline procedure to be one full pass through the data training the likelihood term, and then one full pass through the data training the reconstruction term.

### G.1 Simulated Data

The data for this experiment is simulated from a von Mises distribution centred at $\frac{\pi}{2}$ projected onto a circle of radius 1. We randomly generate 10,000 training data points and train with batch sizes of 1,000. We use 1,000 points for validation, performing early stopping using the value of the full objective and halting training when we do not see any validation improvement for 50 epochs. We create visualizations in Figure 1 by taking 1,000 grid points equally-spaced between $-3$ and 3 as the low-dimensional space, project these into higher dimensions by applying the flow $g_\phi$, and then assign density to these points using the injective change-of-variable formula (2). In this low-dimensional example, we use the full Jacobian-transpose-Jacobian which ends up just being a scalar as $d = 1$. We commence likelihood annealing (when active) on the 500-th training epoch and end up with a full likelihood term by the 1000-th.

For the $D$-dimensional square flow $f_\theta$, we used a 5-layer RealNVP model, with each layer having a fully-connected coupler network of size $2 \times 10$, i.e. 2 hidden layers each of size 10, outputting the shift and (log) scale values. The baseline additionally uses a simple shift-and-scale transformation in $d$-dimensional space as $h_\eta$; we simply use the identity map for $h_\eta$ in this simple example.

We perform slightly different parameter sweeps for the two methods based on preliminary exploration. For the baseline two-step procedure, we perform runs over the following grid:

- Learning rate: $10^{-3}$, $10^{-4}$.
- Regularization parameter ($\beta$): 10, 50, 100, 200, 1,000, 10,000 (which for this method is equivalent to having a separate learning rate for the regularization objective).
- Likelihood annealing: `True` or `False`.

For our method, we search over the following, although noting that our method was stable at the higher learning rate of $10^{-3}$:

- Learning rate: $10^{-3}$, $10^{-4}$.
- Regularization parameter ($\beta$): 10, 50, 200.
- Likelihood annealing: `True` or `False`.

Empirically we found the two-step baseline performed better with the higher regularization, which also agrees with the hyperparameter settings from their paper. Note that we have searched over 2 times as many runs for the baseline and still obtain better runs with our approach.

**Divergences on RNFs-TS between our codebase and the implementation of Brehmer and Cranmer [8]**  Although we were able to replicate the baseline RNF-TS method, there were some different choices made in the codebase of the baseline method (available here: `https://github.com/johannbrehmer/manifold-flow`), which we outline below:

- The baseline was trained for 120 epochs and then selects the model with best validation score, whereas we use early stopping over an (essentially) unlimited number of epochs.
- The baseline weights the reconstruction term with a factor of 100 and the likelihood term with a factor of 0.1. This is equivalent in our codebase to setting $\beta = 1,000$, and lowering the learning rate by a factor of 10.
- The baseline uses cosine annealing of the learning rate, which we do not use.
- The baseline includes a sharp Normal base distribution on the pulled-back padded coordinates. We neglected to include this as it isn't mentioned in the paper and can end up resulting in essentially a square flow construction.
- The baseline uses the ADAMW optimizer [38] to fix issues with weight decay within ADAM (which they also use). We stick with standard ADAM as we do not use weight decay.
- The baseline flow reparametrizes the scale $s$ of the RealNVP network as $s = \sigma(\tilde{s}+2)+10^{-3}$, where $\tilde{s}$ is the unconstrained scale and $\sigma$ is the sigmoid function, but this constrains the scale to be less than $1 + 10^{-3}$. This appears to be done for stability of the transformation (cf. the ResNets below). We instead use the standard parametrization of $s = \exp(\tilde{s})$ as the fully-connected networks appear to be adequately stable.
- The Baseline uses ResNets with ReLU activation of size $2 \times 100$ as the affine coupling networks. We use MLPs with tanh activation function instead.
- The baseline uses a dataset which is not strictly on a manifold. The radius of a point on the circle is sampled from $\mathcal{N}(1, 0.01^2)$. We use a strictly one-dimensional distribution instead with a von Mises distribution on the angle as noted above.

In general, we favoured more standard and simpler choices for modelling the circle, outside of the likelihood annealing which is non-standard.

**Densities of *all* runs**  We note that, while the results reported in the main manuscript are representative of common runs, both for RNFs-ML (exact) and RNFs-TS; not every single run of RNFs-ML (exact) obtained results as good as the ones from the main manuscript. Similarly, some runs of RNFs-TS recovered better likelihoods than the one from the main manuscript. We emphasize again that the results reported on the main manuscript are the most common ones: most RNFs-ML (exact) runs correctly recovered both the manifold and the distribution on it, and most RNFs-TS runs recovered only the manifold correctly. For completeness, we include in Figures 3, 4, and 5 all the runs we

obtained, where it becomes evident that RNFs-ML consistently outperforms RNFs-TS across runs and hyperparameter values. However we do note in Figure 3b that we get the perpendicular effect that Brehmer and Cranmer [8] predicted might happen if optimizing the full objective, although this is far from typical of our results.

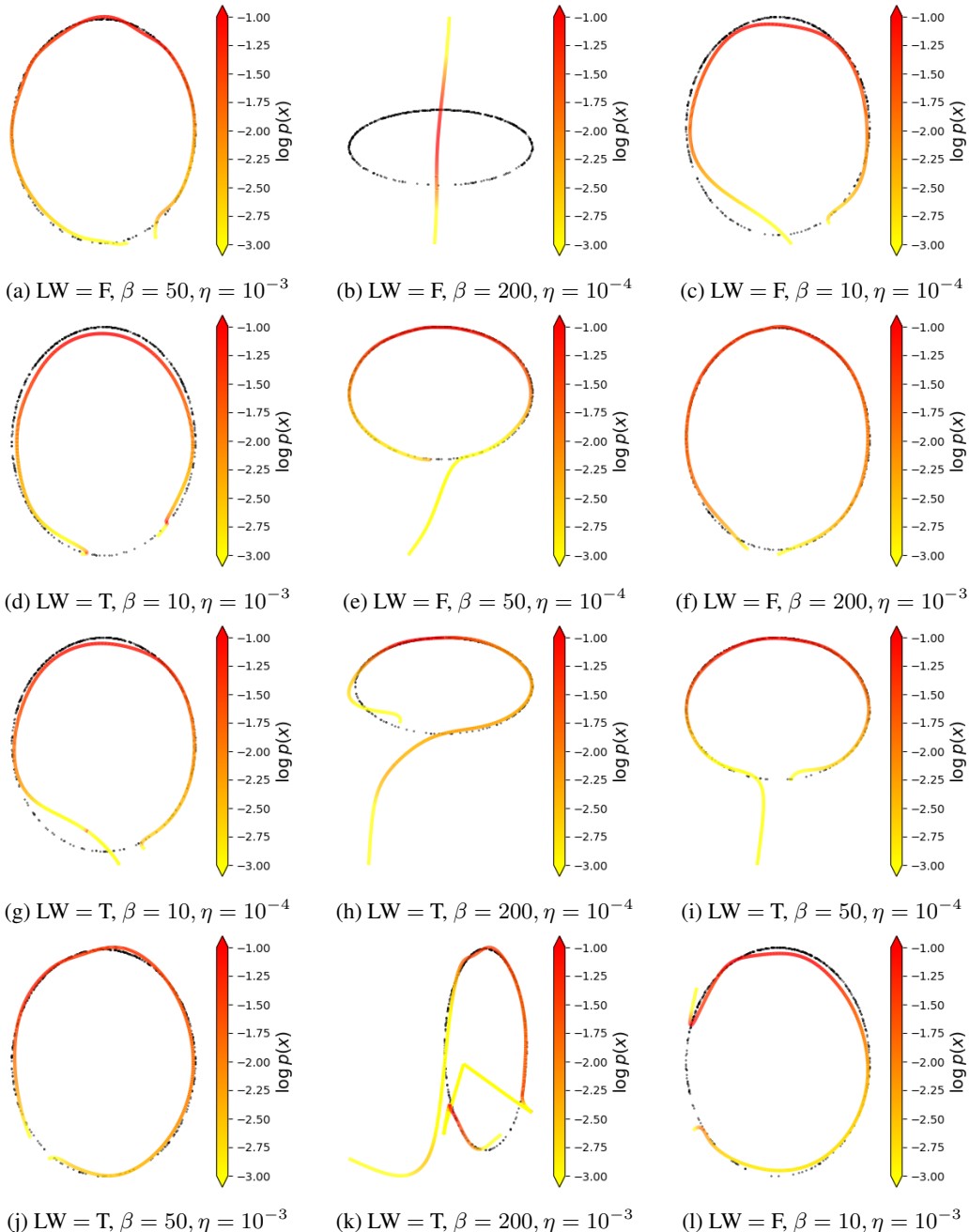

Figure 3: Runs of RNFs-ML (exact), swept over the hyperparameter combinations $\{\texttt{Likelihood Warmup} \in \{\texttt{True}, \texttt{False}\}\} \times \{\beta \in \{10, 50, 200\}\} \times \{\eta \in \{10^{-3}, 10^{-4}\}\}$

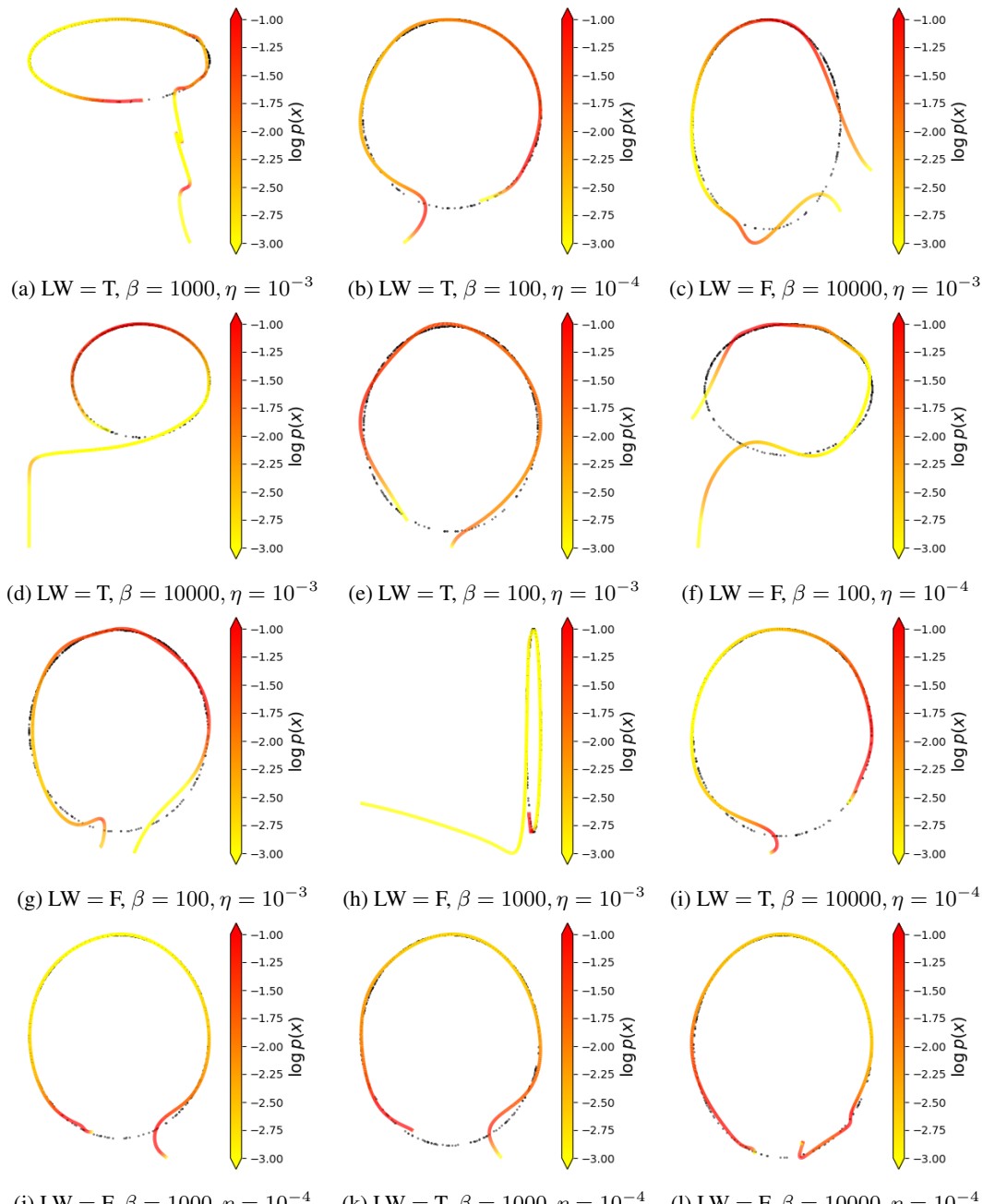

Figure 4: Runs of RNFs-TS, swept over the hyperparameter combinations {Likelihood Warmup ∈ {True, False}} × {$\beta \in \{100, 1000, 10000\}$} × {$\eta \in \{10^{-3}, 10^{-4}\}$}

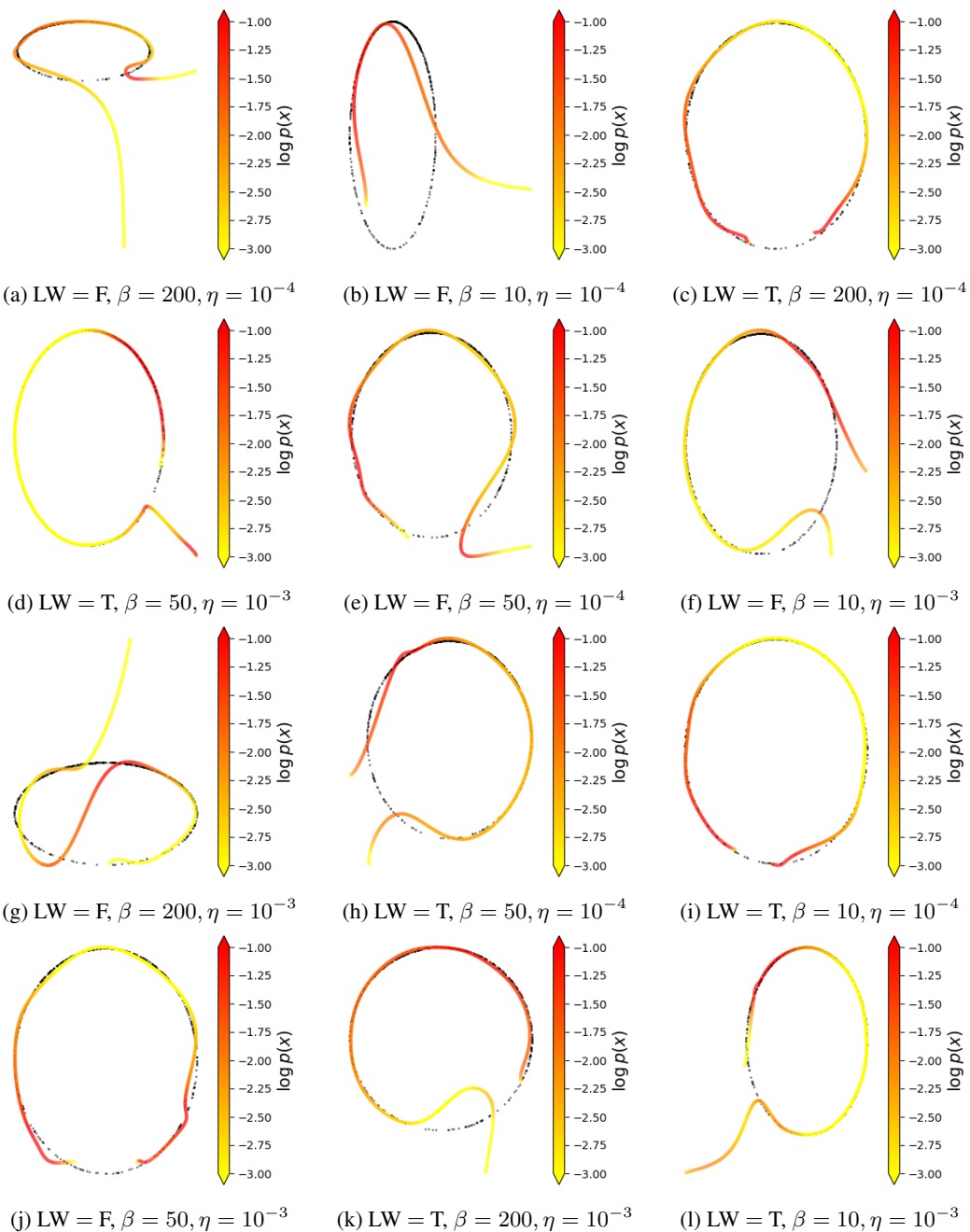

Figure 5: Runs of RNFs-TS, swept over the hyperparameter combinations {Likelihood Warmup $\in$ {True, False}} $\times$ {$\beta \in \{10, 50, 200\}$} $\times$ {$\eta \in \{10^{-3}, 10^{-4}\}$}

## G.2 Tabular Data

For the tabular data, we use the GAS, POWER, HEPMASS, and MINIBOONE datasets, preprocessed as in Papamakarios et al. [45]. We did not observe problems with overfitting in practice for any of the methods. We use the FID-like metric with the first and second moments of the generated and observed data as described in Appendix F for early stopping, halting training after 20 epochs of no improvement.

We again use a RealNVP flow in $D$ dimensions but now with 10 layers, with each layer having a fully-connected coupler network of hidden dimension $4 \times 128$. The $d$-dimensional flow here is also a RealNVP, but just a 5-layer network with couplers of size $2 \times 32$.

In all methods, we use a regularization parameter of $\beta = 50$. We introduce the likelihood term with low weight after 25 epochs, linearly increasing its contribution to the objective until it is set to its full weight after 50 epochs. We select $d$ as $\lfloor \frac{D}{2} \rfloor$, except for ML methods on $D = 8$ GAS which use $d = 2$ (noted below). We use a learning rate of $10^{-4}$. For the methods involving the Hutchinson estimator, we use a standard Gaussian as the estimating distribution. We also experimented with a Rademacher distribution here but found the Gaussian to be superior.

Results reported on the main manuscript are the mean of 5 runs (with different seeds) plus/minus standard error. Occasionally, both RNFs-ML and RNFs-TS resulted in failed runs with FID-like scores at least an order of magnitude larger than other runs. In these rare instances, we did another run and ignored the outlier. We did this for both methods, and we do point out that RNFs-ML did not have a higher number of failed runs.

As mentioned in the main manuscript, GAS required slightly more tuning as RNFs-ML did not outperform RNFs-TS when using $d = 4$. We instead use latent dimension $d = 2$, where this time RNFs-ML did outperform. Since RNFs-TS did better with $d = 4$, we report those numbers in the main manuscript. Otherwise, our methods outperformed the baseline out-of-the-box, using parameter configurations gleaned from the image and circle experiments.

We also include the batch sizes here for completeness, which were set to be reasonably large for the purposes of speeding up the runs:

- POWER - 5,000
- GAS - 2,500
- HEPMASS - 750
- MINIBOONE - 400

### G.3 Image Data and Out-of-Distribution Detection

In this set of experiments, we mostly tuned the RNFs-ML methods on MNIST for $K = 1$ – applying any applicable settings to RNFs-TS on MNIST as well – which is likely one of the main reasons that RNFs-ML perform so well for $K = 1$ vs. the exact method or $K = 4$. The reason why we spent so much time on $K = 1$ is that it was the fastest experiment to run and thus the easiest to iterate on. Our general strategy for tuning was to stick to a base set of parameters that performed reasonably well and then try various things to improve performance. A full grid search of all the parameters we might have wanted to try was quite prohibitive on the compute that we had available. Some specific details on settings follow below.

For the $D$-dimensional square flow, we mainly used the 10-layer RealNVP model which exactly mirrors the setup that Dinh et al. [15] used on image data, except we neglect to include batch normalization (as discussed in Appendix E) and we also tried reducing the size of the ResNet coupling networks from $8 \times 64$ to $4 \times 64$ for computational purposes. For further computational savings, we additionally attempted to use a RealNVP with fewer layers as the $D$-dimensional square flow, but this performed extremely poorly and we did not revisit it. For the $d$-dimensional square component, we used another RealNVP with either 5 or 10 layers, and fully-connected coupler networks of size $4 \times 32$. We also looked into modifying the flow here to be a neural spline flow [16], but this, like the smaller $D$-dimensional RealNVP, performed very poorly as well. This may be because we did not constrain the norm of the gradients, although further investigation is required. We also looked into using no $d$-dimensional flow for our methods as in the circle experiment, but this did not work well at all.

For padding, we first randomly (although this is fixed once the run begins) permute the $d$-dimensional input, pad to get to the appropriate length of vector, and then reshape to put into image dimension. We also pad with zeros when performing the inverse of the density split operation (cf. the $z$ to $x$ direction of Dinh et al. [15, Figure 4(b)]), so that the input is actually padded twice at various steps of the flow.

When we used likelihood annealing, we did the same thing as for the tabular data: optimize only the reconstruction term for 25 epochs, then slowly and linearly introduce the likelihood term up until it has a weight of 1 in the objective function after epoch 50.

We summarize our attempted parameters in Table 5. For some choices of parameters, such as likelihood annealing set to `False`, $d = 15, 30$, $\beta = 10{,}000$, and CG tolerance set to 1, we had very few runs because of computational reasons. However, we note that the run with low CG tolerance ends up being the most successful run on MNIST. We have included "SHORT NAMES" in the table for ease of listing hyperparameter values for the runs in Table 3, which we now provide for MNIST and FMNIST in Table 6 and Table 7 respectively. We also include batch sizes in the table. Note that $^*$ indicates that the run was launched on 2 GPU cards simultaneously, whereas $^{**}$ indicates that the run was launched on 4 GPU cards. For the CIFAR-10 parameters, we attempt several runs of the best configurations below: we sent 1 run for RNFs-ML (exact), 2 runs for RNFs-ML ($K = 1$), and 4 runs for RNFs-TS.

Table 5: Parameter combinations investigated for MNIST runs. Note that the final two rows are irrelevant for RNF-ML (exact) and RNF-TS. We include "short names" for ease of listing parameters for the runs in Table 3.

| PARAMETER | SHORT NAME | MAIN VALUE | ALTERNATIVES |
|---|---|---|---|
| Likelihood Annealing | LA | True | False |
| Reconstruction parameter | $\beta$ | 50 | $5, 500, 10000$ |
| Low dimension | $d$ | 20 | $10, 15, 30$ |
| $D$-dim flow coupler | $D$ NET | $8 \times 64$ | $4 \times 64$ |
| $d$-dim flow layers | $d$ LAYERS | 5 | 10 |
| Hutchinson distribution | HUTCH | Gaussian | Rademacher |
| CG tolerance (normalized) | tol | 1 | 0.001 |

Table 6: Parameter choices for the MNIST runs reported in Table 3.

| METHOD | LA | $\beta$ | $d$ | $D$ NET | $d$ LAYERS | HUTCH | tol | BATCH |
|---|---|---|---|---|---|---|---|---|
| RNFs-ML (exact) | True | 5 | 20 | $8 \times 64$ | 10 | N/A | N/A | 100** |
| RNFs-ML ($K = 1$) | True | 5 | 20 | $8 \times 64$ | 10 | Gaussian | 0.001 | 200 |
| RNFs-ML ($K = 4$) | True | 50 | 20 | $8 \times 64$ | 5 | Gaussian | 1 | 100* |
| RNFs-TS | True | 50 | 20 | $8 \times 64$ | 5 | N/A | N/A | 200 |

Table 7: Parameter choices for the FMNIST runs reported in Table 3.

| METHOD | LA | $\beta$ | $d$ | $D$ NET | $d$ LAYERS | HUTCH | tol | BATCH |
|---|---|---|---|---|---|---|---|---|
| RNFs-ML (exact) | True | 50 | 20 | $8 \times 64$ | 10 | N/A | N/A | 100** |
| RNFs-ML ($K = 1$) | True | 50 | 20 | $8 \times 64$ | 5 | Rademacher | 1 | 200 |
| RNFs-ML ($K = 4$) | True | 50 | 20 | $8 \times 64$ | 10 | Rademacher | 1 | 200** |
| RNFs-TS | False | 5 | 20 | $4 \times 64$ | 10 | N/A | N/A | 200 |

**Visualizations** As an attempt to visualize the learned manifold, we also trained our model with $d = 2$, and show the samples obtained for different values of $z \in \mathbb{R}^2$ in Figure 6, where the spatial location of each sample is given by the Cartesian coordinates of the corresponding $z$ value. While there are some abrupt changes, which we believe are to be expected since the true manifold likely consists of several connected components, we can see that for the most part similar-looking images have nearby latent representations.

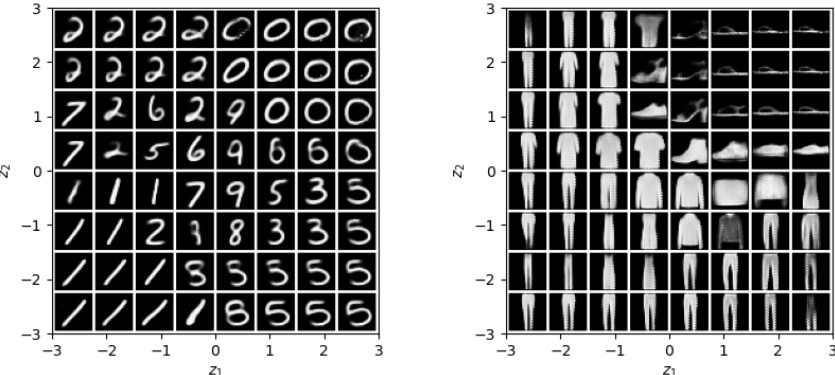

Figure 6: MNIST (left) and FMNIST (right) samples from RNFs-ML (exact) with $d = 2$. The subindices in $z_1$ and $z_2$ index coordinates, not datapoint number.

**Further Out-of-Distribution Detection Results**    Figure 7 shows RNFs-ML log-likelihoods for models trained on MNIST (**left panel**), and we can see that indeed MNIST is assigned higher likelihoods than FMNIST. We also include OoD detection results when using reconstruction error instead of log-likelihoods, for models trained on FMNIST (**middle panel**) and MNIST (**right panel**). We observed similar results with RNFs-TS. Surprisingly, it is now the reconstruction error which exhibits puzzling behaviour: it is *always* lower on FMNIST, regardless of whether the model was trained on FMNIST or MNIST. Once again, this behaviour also happens for RNFs-TS, where the reconstruction error is optimized separately. We thus hypothesize that this behaviour is not due to maximum likelihood training, and rather is a consequence of inductive biases of the architecture.

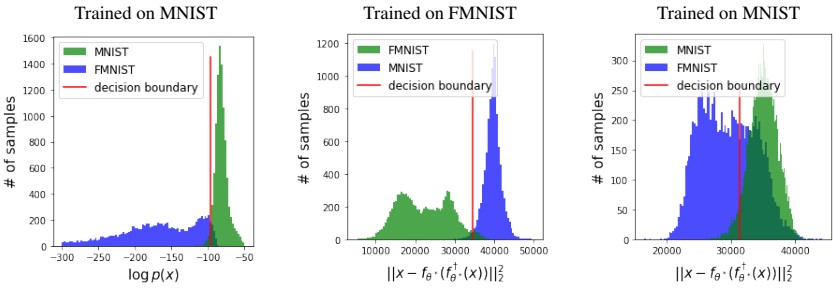

Figure 7: OoD log-likelihood histograms trained on MNIST (left), and OoD reconstruction error histograms trained on FMNIST (middle) and MNIST (right). Log-likelihood results (left) are RNFs-ML (exact), and reconstruction results (middle and right) are RNFs-ML ($K = 1$). Note that green denotes in-distribution data, and blue OoD data; and colors *do not* correspond to datasets.