# OpenReview forum: "Rectangular Flows for Manifold Learning"
_NeurIPS.cc/2021/Conference — NeurIPS 2021 Poster_

### Official Review · Reviewer_b6vS · 2021-07-15

**Rating:** 4
**Confidence:** 4

**Summary:**

The task of interest is to simultaneously learn a manifold (with a fixed predefined dimensionality) and the density on it, from training data. The proposed method, rectangular flows, is developed based on an existing work (Brehmer and Cranmer [6]) that used a two-step training procedure (for manifold matching and density matching, respectively). The main contributions include: (1) revealing that the two-step training can be improved by joint training; (2) a new technique to enable computing the Jacobian-transpose-Jacobian term of the joint training objective with better efficiency. Experiments on simulated data and MNIST/FMNIST are conducted to demonstrate the proposed method.

**Limitations And Societal Impact:**

Yes.

**Main Review:**

Originality. The presented method in Section 4.2, for calculating the volume-change term, is believed new; it's a novel combination of existing techniques.

Quality. The presented method is technically sound. Some empirical observations are not discussed/analyzed in detail. For example, the inconsistent results on simulated data, and the unexpected behavior on MNIST/FMNIST.

Clarity. The writing is OK, but can be certainly improved; for example, to highlight the underlying logic, the main contributions over existing methods, algorithms that show the big picture and/or the details (like VJP/JVP), and so on.

Significance. The contributions might be incremental; the presented method enables joint training for existing work, but the corresponding importance is not well demonstrated with serious experiments and/or analysis.

Other comments:

Line 48, the exact method is quite straightforward. I don't think it is a contribution, so no credit for it.

Eq (8). Since beta is just treated as a hyperparameter, the KKT conditions are not met or used. Please clarify.

Lines 164-165. What's the meaning of the statements here?

How to calculate a VJP or a JVP? It's not clear in the manuscript.

Eq (12). Since the (J^T J eps) term of the loss is calculated via VJP/JVP, how do you guarantee the correct gradient wrt theta?

Line 323. Why RNFs can remedy the phenomenon? Also, too many conjectures are made in this paragraph without solid analysis.


**Time Spent Reviewing:**

3

---

> ### Author Response · Authors · 2021-08-09
> **Author Response**
>
> Thank you for your review. Please see points 1, 2, and 3 in our general rebuttal for answers to several of the points you raised. We address the rest of your points below:
> 1. On novelty: As mentioned in point 1 of our general rebuttal, we believe that part of the method and thus its novelty might have been misunderstood, particularly given some of the additional questions about jvps and vjps (which we address below). The novelty of the exact method does not come from doing an exact computation, rather it stems from \textit{how} this exact computation is carried out. Naively using backpropagation (vjps) to construct the Jacobian for the exact computation is unnecessarily expensive, and we show that using forward-mode AD (jvps) -- which is not the default, nor even efficiently implemented in PyTorch -- results in the same exact computation being performed much faster.
> 2. On AD methods not being clearly explained: first, we point out that we actually have a brief overview of AD, both forward- and backward-mode in appendix C, and we also cite [1], a very comprehensive survey on the topic. We compute both vjps and jvps in standard ways; these methods are well established within the literature and we believe that our explanations in appendix C are already more than usually presented when using these methods, and a thorough explanation of them falls outside of the scope of our paper. For vjps we use PyTorch's bakpropagation implementation, and for jvps we use our custom implementation. We achieve this by having layers and nonlinearities not only take $x$ as input, but also a vector $v$, and outputting $Jv$ in addition to the usual output.
> 3. On lines 164-165: We believe that "since our likelihoods are Radon-Nikodym derivatives with respect to the Riemannian measure on $\mathcal{M}_\theta$, different values of $\theta$ will result in different dominating measures" is a mathematically precise and unambiguous statement. The intuitive meaning of this is that the support changes as $\theta$ changes, with the implication being that one should be careful when comparing likelihoods for different values of $\theta$, even within the same architecture (which is why we did not compare against the baseline using likelihoods). A formal explanation of these concepts falls outside the scope of our paper, but we refer the reviewer to [3] for a formal treatment of probability, which includes Radon-Nikodym derivatives, and to [4] for a definition of Riemannian measures.
> 4. On equation (12) recovering the correct gradient: The fact that the $J_\theta^T J_\theta \epsilon$ terms are computed using vjps and jvps (with respect to inputs) does not imply that $J_\theta^T J_\theta \epsilon$ is not computed exactly, or that its gradient (with respect to $\theta$) cannot be computed; it is only specifying _how_ we compute $J_\theta^T J_\theta \epsilon$. Thus, using vjps and jvps does in no way result in obtaining incorrect gradients with respect to $\theta$. Once again, we refer to appendix C and to [1] for AD details.
> 5. On the paragraph starting in line 323: As mentioned in the paper and also pointed out by reviewer __zGxK__, we hypothesize that RNFs can help remedy OoD issues through the correct inductive bias of having the intrinsic data dimension being lower than the ambient dimension. We believe this is a reasonable hypothesis, and is clearly stated as a conjecture, and a theoretical verification falls outside the scope of our paper. We also point out that this is actually the only conjecture we put forward in this paragraph, when we say "this seems to be a property of RNFs rather than our training method" we are only referring to the fact that baseline RNFs-TS also had strong OoD empirical performance. We will clarify this by publication time.
> 6. On the KKT conditions not holding: We point out that the KKT conditions do hold. The constrained optimization objective is:
>
>   $$
> \text{argmax} _\phi \sum _{i=1}^n \left\\{\log p _Z\left(g _\phi^\dagger (x _i )\right)  - \log\left|\det \mathbf J [h _\eta]\left(g _\phi^\dagger(x _i)\right)\right|-\frac{1}{2}\log \det J _\theta^\top(x _i)J _\theta(x _i)\right\\}
>   $$
>
> $\text{subject to} \displaystyle \sum_{i=1}^n \left|\left|x_i - f_\theta\left(f_\theta^\dagger(x_i)\right)\right|\right|_2^2 \leq \lambda,$
> where $\lambda > 0$ is a hyperparameter which implicitly defines $\beta$ from equation (8) in the paper. Rather than treat $\lambda$ as the hyperparameter, we treat $\beta$ directly as the hyperparameter and optimize the Lagrangian (equation (8)), which is a standard use of the KKT conditions in machine learning. We will include this in the appendix, and point to [2] for a thorough treatment of the KKT conditions.
>
> [1] Automatic Differentiation in Machine Learning: A Survey, Baydin et al.
>
> [2] Constrained Optimization and Lagrange Multiplier Methods, Bertsekas
>
> [3] Probability and Measure, Billingsley
>
> [4] Intrinsic Statistics on Riemannian Manifolds: Basic Tools for Geometric Measurements, Pennec

---

### Official Review · Reviewer_Jpwi · 2021-07-16

**Rating:** 7
**Confidence:** 4

**Summary:**

The paper presents new methods for estimating densities using rectangular normalizing flows. Rectangular normalizing flows, unlike squared normalizing flows, allow considering injective instead of bijective flows. This property is exploited for estimating the density of data residing on a low (d) dimensional manifold in R^D, for D>d. Existing works using rectangular flows to estimate densities supported on low dimensional manifolds assume that the manifold is known, and therefore are restricted to tori, spheres, etc. In contrast, this work does not make this assumption and shows that density estimation for injective flows based on ML is tractable. Two methods are proposed, which are based on incorporating and estimating the “Jacobian-transpose-Jacobian” in the objective. One is based on an exact derivation, which is computationally demanding. The other is based on stochastic gradient estimates. Experimental results on a simulation and a couple of datasets are shown, demonstrating the advantage of the proposed methods compared to an existing method as well as the tradeoff between memory and variance incurred by the two proposed methods.

**Ethical Concerns:**

I do not have any ethical concerns.

**Limitations And Societal Impact:**

The authors very nicely discuss the limitations of their work.
I do not foresee any potential negative societal impact

**Main Review:**

This is a well-written, timely, and interesting paper. It addresses an important problem and presents a solid contribution.
The work has several notable limitations:
- It can only learn densities supported on manifolds homeomorphic to R^d.
- It is computationally expensive and feasible only when the manifold dimension is low.
- Partly as a result of the limitation above, the experimental study is not impressive as it includes only relatively simple examples.
I commend the authors for clearly putting these limitations forward in a nice discussion in Section 6, which also highlights some intriguing results and the gained insights.
Despite its shortcomings, I believe the paper improves existing methods and advances this line of work, and I would be happy to see it published.

Minor comments:
- A clear description of the two algorithms would help (even if put in the appendix due to length limitations).
- More details on the choice of the number of gradients to estimate K (in the stochastic variant) would help practitioners.
- In Appendix F, failed runs on the simulated data are presented, which raises two questions: is there an explanation for the failures? can the results be quantified (over many runs)?
- The FID score, which is used to evaluate the results in Sec 5.2, is defined only in the appendix. I believe that some details, or a basic definition, need to appear in the main paper.
- In the experimental study on MNIST (Sec 5.3), the dimension d is set to 20. Why? Was it obtained empirically? Is it based on previous works? Is this a result of the computational limitation? Please elaborate.


**Time Spent Reviewing:**

4

---

> ### Author Response · Authors · 2021-08-09
> **Author Response**
>
> Thank you for your review. Please see point 3 in our general rebuttal for an answer to your question about failed runs on simulated data. We will incorporate all your minor comments in a revised version of the manuscript by publication, and address some of your other points below:
>
> 1. On only learning densities supported on manifolds which are homeomorphic to $\mathbb{R}^d$ being a notable limitation: While we agree this is a limitation of our method and believe this should be addressed in future work (all of which we mentioned in our manuscript), we think it's also fair to highlight that almost all current models suffer from similar topological mismatches. For example, GANs, by mapping $\mathbb{R}^d$ through a continuous function cannot model supports with more than a single connected component (since continuous images of connected sets are connected). Similarly, all VAE and EBM methods we are aware of are $\mathbb{R}^D$-supported, resulting in supports (trivially) homeomorphic to $\mathbb{R}^D$, which we believe is even worse. The only work we are aware of considering these topological pathologies is CIFs [1], which provides a more numerically stable approximation of topologically-misspecified supports over baseline square flows, but nonetheless still results in a support defined over $\mathbb{R}^D$.
> As mentioned in our paper, we believe that potentially integrating CIFs within our framework to alleviate numerical issues arising from modelling the manifold as being homeomorphic to $\mathbb{R}^d$ is a promising direction of future work.
> 2. While indeed having higher manifold dimension makes our method more computationally expensive, most high-dimensional datasets of interest will exhibit strong low-dimensional structure, and often have significantly fewer factors of variation than they do features.
> 3. We treated $d$ as a hyperparameter for MNIST, see Table 5 in the appendix for all the hyperparameters that we considered. In particular, for $d$, we considered 10, 15, 20, and 30. We chose these values based on typically-selected values for the latent dimension of VAEs, which we believe are an adequate proxy for the number of factors of variation, or intrinsic dimension, of the data. We will clarify this in the main manuscript.
>
> [1] Relaxing Bijectivity Constraints with Continuously Indexed Normalizing Flows, Cornish et al.

---

> > ### Comment · Reviewer_Jpwi · 2021-09-01
> > **After rebuttal**
> >
> > I thank the authors for the detailed response to my comments.
> > After reading the other reviews and the authors' rebuttal, I am keeping my score and recommend accepting the paper, provided that the authors will incorporate the changes in the revision as they stated in the rebuttal.

---

### Official Review · Reviewer_1ptY · 2021-07-16

**Rating:** 6
**Confidence:** 4

**Summary:**

This paper is motivated by the desire to circumvent a problematic requirement in Normalizing Flows (NFs) on the function $f: \mathbb{R}^d \rightarrow \mathbb{R}^D$ that produces the pushforward measure --- namely, that $f$ must be strictly invertible. This is difficult for generative models, since the dimension of the latent space $d$ is usually much smaller than that of the target data space $D$, i.e. $d << D$. The invertibility requirement centers around the change-of-variables formula, which is crucial for computing the cost function in NFs, and which requires computing the Jacobian of $f^{-1}$. To enable the use of NFs on latent/data spaces where $d << D$, the authors propose "Rectangular" NFs (RNFs). The name comes from the fact that the Jacobian of $f$ is now rectangular in the $d << D$ case. This involves the use of a differential geometric variant of the change-of-variables formula, and is much more computationally taxing, but provides superior density estimation performance to previous work.


**Limitations And Societal Impact:**

There is limited experimental validation of the proposed technique. The authors compare their method to only one other work, on three datasets: a synthetic circle dataset, MNIST, and FMNIST (they mention having tried CIFAR10 and SVHN, but were unable to tune their model in a reasonable amount of time). Even with MNIST and FMNIST, it could be informative to train an NF with a 2-dimensional latent space and visualize manifold to see if some structure of the dataset were preserved.


**Main Review:**

The authors explore a very natural and compelling question that arises from the invertibility requirement (of the aforementioned $f$) in NFs, when applying NFs to generative modeling.

$f$, which can be thought of as the "decoder" or "generator" network, often maps from some lower dimensional latent space to a higher dimensional data space. If $f$ and the latent space are suitably well behaved, it would be reasonable to assume that, even though $f$ is not invertible, its restriction to the data manifold $\mathcal{M}$ could be. Intuitively, if $f$ is injective, this suggests that there does still exist a tractable way to compute the pushforward density, and one wonders if there is a more general case of the change of variables formula that can be used.

To this end the authors demonstrate that, while this general form does exist, the computation of its terms (namely, the inverse of the Jacobian) is unfortunately quite heavy and numerically unstable, and moreover, stochastic estimations of this Jacobian via random projections is noisy.

Though these are not strictly positive results, they highlight important obstacles to further progress in an area of very active research. Moreover, the work is presented in a lucid and understandable manner, and the limitations of the current method are clearly delineated. For these reasons, I am inclined to accept the paper.

**Time Spent Reviewing:**

6

---

> ### Author Response · Authors · 2021-08-09
> **Author Response**
>
> Thank you for your review. Please see point 2 in our general rebuttal for answers to some of your experimental concerns. Could you please elaborate on what is meant by our results being "not strcitly positive"? We believe we have improved over the two-step baseline in every dataset we tried.

---

### Official Review · Reviewer_zGxK · 2021-07-17

**Rating:** 7
**Confidence:** 4

**Summary:**

This paper addresses the problem of constructing injective normalizing flows, which connect some low-dimensional space with the data manifold of interest in the high-dimensional space. Typically injective (or rectangular) flows are composed of two square flows with the “upsampling” padding layer between them. Because the volume change term in the case of injective flows is seemingly intractable, current approaches find workarounds by training these two square flows separately step-by-step. Though this decision has certain drawbacks and may result in suboptimal performance. Current work suggests to employ numerical linear algebra methods to make volume change computation tractable. Experiments demonstrate that taking into account volume сhanging term improves the generation results both for synthetic and real-life data. Interestingly, it also improves out of distribution detection score, allowing the model trained on FashionMNItST to assign lower probability for the MNIST data samples.

**Limitations And Societal Impact:**

The authors adequately addressed the limitations.

**Main Review:**

- Paper is clearly written, exposition is coherent with all the  preliminaries and motivations for the technical decisions in place. Derivations are plain and comprehensible.

- While injective flows as a class were described before, previous heuristics for implementing them were oversimplified and in the best case suboptimal. Thus making volume-change tractable for the injective flows seems to be an important step forward.

- While (as stated in the paper) conjugate gradients were previously applied to the calculation of the volume change for the square flows, authors are the first to adopt them to the rectangular flows and Jacobian-Transpose Jacobian product calculation. Thus the novelty in this sense is moderate. Any comments?

- Experiments on the synthetic and tabular data are convincing and clear demonstrate, that two-step method leads to learning incorrect probability distribution. On the Fashion MNIST FID improvements are quite notable.

- Though, for me, the fact that increasing the number of steps and thus decreasing variance leads to worse results is a bit puzzling, and thus I would like to see more experiments in this direction, which would clarify this question.  Any comments?

- Out-of-distribution experiments also are promising and hint that low-dimensional manifold is a important inductive bias.

- Overall the result on the toy data and simple image data are quite convincing, though until it is verified on the real-life level of complexity data the broader significance is under question. Thus I would also l like to see some experiments on CIFAR to get sure.  Any comments?

**Time Spent Reviewing:**

2.0

---

> ### Author Response · Authors · 2021-08-09
> **Author Response**
>
> Thank you for your review. Please see points 1 and 2 in our general rebuttal for answers to several of the points you raised. As to why increasing $K$ led to worse results, we first clarify that $K$ is the number of Hutchinson samples used in our stochastic estimator, and not the number of flow steps, in case this caused confusion (although perhaps this was not what was meant by "increasing number of steps"). We do explain this in line 314, but agree that it warrants more emphasis and will add it by publication: since we used $K=1$ to tune hyperparameters (providing the baseline with the same amount of tuning)
> and then used the same hyperparameters for $K=4$ and the exact method, the selected hyperparameters seem to favour $K=1$.

---

> > ### Comment · Reviewer_zGxK · 2021-09-02
> > **Comment to Author Response**
> >
> > The authors addressed my comments. I think that despite the drawbacks mentioned by other reviewers, the paper has practical merits, and can be accepted. I decided to increase my score.

---

### Official Review · Reviewer_mod8 · 2021-07-18

**Rating:** 6
**Confidence:** 3

**Summary:**

The paper addresses the challenge of learning a normalizing flow, under the assumption that the data probability function is concentrated on a low dimensional manifold. Utilizing the manifold assumption introduces a challenge in computing the volume change term. This paper suggests addressing this computational challenge in two ways: i) If the manifold dimension is low enough, it is reasonable to calculate the map differential exactly based on AD forward mode; ii) alternatively, an unbiased stochastic estimator for the log det can be used.


**Limitations And Societal Impact:**

Yes

**Main Review:**

The paper addresses the challenge of learning normalizing flows. Previous work has recently suggested incorporating the manifold assumption into the normalizing flow [1].  The loss that follows from the manifold normalizing flows in [1] depends on a volume change term, a term that [1] did not compute directly. The current paper claims that there is an advantage in optimizing directly the loss involving the volume change term, and suggests two options to compute it. Even though there is not a lot of novelty in the techniques related to the suggested computations, I mark this paper as a solid contribution to the ongoing effort of learning normalizing flows and rate it with a positive rating. Detailed comments are provided next.

Generally, the paper is well written and easy to follow. The introduction and the method presentation are of good quality. I appreciate the author's effort to include all the relevant details to the method, in addition to some issues that were considered in devising the method. This is also reflected in the experiments section, where details about failed experiments are included.

The main contribution of the paper is that in fact the loss suggested in [1] can be computed directly in some cases or stochastically estimated. It seems to me that the stochastic estimation of the logdet is not novel, see [2]. Also, the exact computation is based on the forward mode AD, which is a known technique.

I am concerned about the inconsistency in the results of figure 1. Do the authors have some thoughts on what can be done? This should be in the main text.

The experiments show an advantage over [1], except for the OOD experiment. However, why other alternatives have not been considered? In table 3 it would be important to see results from other square flow models (without the manifold assumption), or contractive auto-encoders (that are known to approximate the derivative of the data log probability function [3]). As the current method cannot run on more complex datasets like CIFAR, these additional comparisons can some light if this line of works is worth pursuing.

[1] : Flows for simultaneous manifold learning and density estimation, Brehmer and  Cranmer.
[2] : Invertible Residual Networks, Behrmann et al.
[3] : What Regularized Auto-Encoders Learn from the Data Generating Distribution, Alain and Bengio,

**Time Spent Reviewing:**

5

---

> ### Author Response · Authors · 2021-08-09
> **Author Response**
>
> Thank you for your review. Please see points 1, 2 and 3 in our general rebuttal for answers to most of the points you raised. We would like to highlight that, while the baseline performed better than our method for OoD detection on FMNIST/MNIST, the results were close, and we used our exact, forward-mode AD-based method to compute log likelihoods for the baseline at test time. Thus, the baseline results for OoD detection are only realizable thanks to our contributions (indeed, [1] does not report OoD results).
>
> [1] Flows for Simultaneous Manifold Learning and Density Estimation, Brehmer and Cranmer

---

### Author Response · Authors · 2021-08-09
**General Response to All Reviewers**

We thank the reviewers for their comments and feedback on our work, as well as for finding it "well-written, timely, and interesting" (__Jpwi__), an "important step forward" (__zGxK__), and a "solid contribution" (__mod8__ and __Jpwi__). Below we address high-level points and questions shared by several reviewers, and answer more specific points to each reviewer individually.

1. __On novelty__ (reviewers __mod8__, __zGxK__, __b6vS__): We respectfully contest claims that our only novelty is the application and not our logdet (and logdet's gradient) estimators. As we point out in the paper, logdet approximations and CG have been used before in the context of NFs, but not in the context of rectangular flows, which require additional contribution to be efficient. For example, our solution involving forward-mode AD results in the first reported experiments where the logdet term is included in the objective of injective flows. For a further example, the paper mentioned by reviewer __mod8__ [1] (which we will cite and discuss in the paper) does indeed approximate a logdet, but uses power series -- which as discussed in our paper, results in biased estimates -- and does not require the use of forward-mode AD, in addition to only being applied to square flows. We stand behind our claim that the careful use of AD, both for our exact and stochastic methods, is novel in the context of rectangular flows. In particular, we believe reviewer __b6vS__'s comment that the exact method is straightforward might come from a misunderstanding about how we use forward-mode AD (see the individual answer to reviewer __b6vS__ for more detail).
2. __On experimental choices and results__ (reviewers __mod8__, __zGxK__, __1ptY__, __b6vS__): We have only compared against the method of [2] since our method is designed to train the exact same model with an improved objective. Fair comparisons are thus straightforward as the exact same architecture can be used, thus allowing to see the effect of changing the objective. Comparing against, for example [3] as suggested by reviewer __mod8__, would confound comparisons and thus not allow to compare the objectives themselves.
Furthermore, the comment by reviewer __1ptY__ that we only compared on three datasets has neglected to acknowledge the four tabular datasets in section 5.2.
That being said, we thank the reviewers for their suggestions and agree there is some instructive value in other comparisons, and will include comparisons against square flows by publication time if asked by the reviewers, although we also note that [2] already showed improved performance of rectangular flows (with two-step training, over which we improve upon) over square flows.
Additionally, we have run our method on CIFAR-10 since submission, and found that our best configuration (using the exact method) outperformed the two-step baseline's best configuration in FID score, $643$ vs $731$ (as in our other comparisons, the same architectures are used and both methods receive comparable amounts of tuning).
Also, as per reviewer __1ptY__'s suggestion, we have ran $d=2$ and plotted the manifold and observed a clear structure of similar images being close-by. We cannot share images in the rebuttal, but will also include this by publication time, along with the CIFAR-10 results.
Finally, reviewer __b6vS__ mentions "unexpected behavior on MNIST/FMNIST" and our results show a convincing improvement on both of those datasets; perhaps they can elaborate on what this behaviour is?
3. __On perceived inconsistencies of Figure 1__ (reviewers __mod8__, __Jpwi__, __b6vS__): We do not believe that the results from Figure 1 are particularly inconsistent, neither for our method nor for the baseline.
Neural networks have complicated optimization landscapes and sometimes fail, and we included failed runs in the appendix in the interest of scientific integrity, even if they are not representative of most runs. Of the $24$ runs we had for each method, ours correctly recovered the manifold $17$ times, and the correct density on it $12$ times (we visually assessed correctness); whereas the baseline recovered the correct manifold $15$ times and the right distribution on it _only_ $2$ times. Additionally, these $24$ runs were done over different hyperparameters (the same hyperparameters were considered for both methods), and it is thus to be expected that some of these are just bad hyperparameter choices.
Additionally, these failures were even rarer in other experiments. We will clarify this point in the manuscript by publication time, and will happily include all these runs in the appendix if requested by the reviewers.

[1] Invertible Residual Networks, Behrmann et al.

[2] Flows for Simultaneous Manifold Learning and Density Estimation, Brehmer and Cranmer

[3] What Regularized Auto-Encoders Learn from the Data Generating Distribution, Alain and Bengio

---

### Author Response · Authors · 2021-08-19
**Rebuttal reminder**

We kindly remind the reviewers about our rebuttal, in which we believe we have addressed their main concerns, including results on CIFAR-10 and clarifying perceived inconsistencies of Figure 1. Please let us know if there are any additional questions, comments, or updated views on our work.

---

### Decision · Program_Chairs · 2021-09-27

**Decision:**

Accept (Poster)

**Comment:**

Congratulations on the acceptance of your paper! Please incorporate changes, edits and additional promised experiments from "Author Discussion" in the final paper/appendix.